# MulCLIP: A Multi-level Alignment Framework for Enhancing Fine-grained Long-context CLIP

## Abstract

Pioneering vision–language models such as CLIP have transformed multimodal learning by aligning images and text in a shared embedding space. However, CLIP's training on short captions limits its ability to handle downstream tasks that require longer text comprehension and fine-grained visual grounding. Recent advances mitigate this challenge by leveraging region-proposal information to map visual regions with corresponding sentences from lengthy captions, yet incurring notable deployment costs. In this paper, we introduce **MulCLIP**, a novel end-to-end multi-level alignment framework that bridges long-text structures (**long captions, sentences, words**) with image components (**global, regional**), enabling fine-grained capabilities while surpassing CLIP's strength on short-text understanding. MulCLIP first preserves global contrastive alignment between images and both summary and long captions, while extending positional embeddings for longer text sequences. To further enhance fine-grained understanding, we propose two novel strategies: (1) a token reconstruction alignment over locally calibrated features to strengthen semantic connections between words and image patches, and (2) a subcaption–aggregated patch alignment that automatically extracts and aggregate context-rich patches for each subcaption. Experimental results demonstrate MulCLIP outperforms baselines in both long- and short-text understanding, while ablation studies confirm its multi-scale alignment is the key factor driving better fine-grained capability than region-proposal–assisted approaches.

## 1 Introduction

Efforts to bridge the alignment gap between visual and linguistic modalities have prominently highlighted the CLIP model (Radford et al., 2021), a multimodal embedding framework trained via contrastive learning on more than 400 million image–text pairs. CLIP effectively maps visual and textual inputs into a shared representation space, showcasing impressive zero-shot generalization across a wide range of downstream tasks including image retrieval, visual question answering, and image captioning. Despite CLIP's strong generalization ability, it remains limited in fine-grained understanding, particularly in recognizing object attributes and their relationships (Wu et al., 2024; Tong et al., 2024). This stems from two key factors: (i) although CLIP's text encoder can process up to 77 tokens, the model is predominantly trained on short, generic captions that emphasize high-level semantics and lack detailed descriptions; and (ii) it performs global alignment between full images and texts, making it difficult to associate localized visual regions with specific textual components. These constraints hinder the model's ability to handle complex scenes and long-form descriptions, where nuanced alignment is essential.

To address such issue, LongCLIP (Zhang et al., 2024) extend CLIP's capacity for long-text modeling by modifying its positional encodings, enabling the model to process longer sequences without disrupting the alignment learned from pre-trained CLIP weights. While effective, they still operate at a global representation level and fail to capture the fine-grained correspondences that naturally arise in detailed descriptions. FineLIP (Asokan et al., 2025) narrows this gap by introducing specialized token-alignment mechanisms between image embeddings and long text embeddings. However, it focus solely on long captions, overlooking the semantically rich short phrases describing specific image regions (Onoe et al., 2024; Urbanek et al., 2024). In addition, training with only long caption

| Long-text Tuning Methods | Long Caption | Short Caption | Word | Region-Proposal-Assisted |
|---|:---:|:---:|:---:|:---:|
| LongCLIPZhang et al. (2024) | ✓ | ✓ | ✗ | ✗ |
| FineLIP Asokan et al. (2025) | ✓ | ✗ | ✓ | ✗ |
| GOAL Choi et al. (2025) | ✓ | ✓ | ✗ | ✓ |
| FG-CLIP(Xie et al., 2025) | ✓ | ✓ | ✗ | ✓ |
| **MulCLIP (ours)** | ✓ | ✓ | ✓ | ✗ |

Table 1: Comparison of components aligned with image features across methods.

leads to degradation in understanding short text, as demonstrated in prior findings of Wu et al. (2024).

GOAL (Choi et al., 2025) tackles both long and short captions with a global–local alignment framework. While it achieves strong fine-grained results and maintains solid zero-shot performance, it relies heavily on external segmentation tools (e.g., SAM (Kirillov et al., 2023)) and post-hoc filtering, adding computation and limiting deployment flexibility. Likewise, FG-CLIP (Xie et al., 2025) focuses on fine-grained pre-training on large-scale data, leveraging YOLO-World (Cheng et al., 2024) region proposals and hard-negative mining. As summarized in Table 1, these approaches differ in the textual granularities they align with image features and in their reliance on region-based modules to locate fine-grained visual components.

In this paper, we introduce **MulCLIP**, a simple yet effective adaptation framework for multi-level image–long text alignment. Unlike existing approaches that focus at most two granularities or region proposals and filtering, MulCLIP employ token reconstruction and sub-aggregated patch mechanism on top of semantic features to further refine them while jointly modeling (i) global-to-global relationships between full images and corresponding long and summary short captions, (ii) local-to-local correspondences between image patches and word embeddings, and (iii) sub-caption-to-image-patches alignments, enabling richer and more precise cross-modal understanding. Our main contributions are summarized as follows:

- We propose a unified multi-level alignment framework that bridging the gap between long-form descriptions and complex visual content at three different scales.

- We conduct comprehensive experiments on a range of cross-modal retrieval benchmarks, demonstrating that MulCLIP outperforms existing leading methods on both lengthy fine-grained and standard retrieval tasks.

- We provide extensive ablations and qualitative analysis to elucidate the impact of each component in our framework, highlighting the advantages of our approach for fine-grained multimodal understanding.

## 2 RELATED WORK

**Vision-Language Models (VLMs).** Contrastive learning has established itself as a leading paradigm for multimodal pre-training, significantly advancing the field of image-text alignment. The pioneering work of CLIP (Radford et al., 2021), employing a dual-encoder architecture trained contrastively on approximately 400 million image–caption pairs, demonstrates robust zero-shot transfer capabilities across various downstream tasks such as open-vocabulary recognition, object detection, and semantic segmentation. Moreover, CLIP has become an essential component in numerous generative vision–language systems, including multimodal language models like LLaVA (Liu et al., 2023) and diffusion models (Nichol et al., 2022; Rombach et al., 2022). Following CLIP success, the next VLM foundation models train on hundred million to billions image-text pairs dataset (Jia et al., 2021; Li et al., 2022) and this trend also propagates into domain-specific VLMs, such as medical imaging application (Zhang et al., 2025). However, these models typically rely on short, broad image descriptions as captions, causing them to miss crucial local-level detailed information.

**Fine-grained understanding in VLMs.** To address these limitations, recent work has shifted towards fine-grained attributes in long text. Some approaches integrate the inherent short descriptions from synthetic long text to vision-language models and retrained it from scratch (Zheng et al., 2024; Wu et al., 2024; Xiao et al., 2025), but this forfeits the rich knowledge of pre-trained models like CLIP, demands large-scale data and computation. CLOC (Chen et al., 2025) takes a different route: it mines two billion image–text pairs, then employs open-vocabulary detectors to align local objects

with phrase-level descriptions, achieving strong localization at the cost of heavy data collection and detector inference.

An alternative, more efficient approach involves fine-tuning existing pre-trained CLIP model. Early works (Huang et al., 2021; Bica et al., 2024), highlight token-level alignment between image patches and text word embeddings, pushing the boundaries of fine-grained image–text understanding. In an emerged direction, LongCLIP (Zhang et al., 2024) or TULIP (Najdenkoska et al., 2024) extends the token capacity of CLIP's text encoder, enabling it to process and represent longer, more descriptive captions. In addition, several dense, detailed image-caption datasets such as DCI (Urbanek et al., 2024) and DOCCI (Onoe et al., 2024) have been introduced, leveraging large vision–language models (LVLMs) to generate fine-grained sub-captions that describe local visual details.

Recent methods, including GOAL (Choi et al., 2025) and FG-CLIP (Xie et al., 2025), exploit these annotations by employing external segmentation tools for explicit region-level alignment. Specifically, GOAL uses SAM to segment images and matches sub-captions with relevant regions via CLIP-based filtering. It then jointly aligns both the full and segmented images with long and sub-captions via unified learning objectives. FG-CLIP adopts a two-stage training strategy: in the first stage, it finetuned on billions pairs to adapt a dual-head CLIP on long and short captions; in the second, it continues training on millions of hard negative caption–image pairs and incorporates grounding information from YOLO to achieve finer-grained understanding. FineLIP (Asokan et al., 2025) adopt refinement modules for both CLIP branches followed by cross-modal late interaction to achieve better alignment between image and long text tokens. However, all of these approaches are either non-unified or address at most two granularities, leaving the gap of unified and effective alignment strategy for fine-grained long-context learning.

## 3 METHOD

### 3.1 GLOBAL-LEVEL ALIGNMENT.

MulCLIP aligns images with both summary short and long captions at global level by leveraging the global token embeddings produced by respective visual and textual encoders. To handle text sequences longer than CLIP's standard 77 token limit, we adopt LongCLIP's positional embedding interpolation strategy in our text encoder. This adjustment allows longer text inputs while minimizing disruptions to the strong crossmodal alignment achieved in the pretrained CLIP.

Formally, consider a CLIP-style vision–language model $f = (f_v, f_v^h, f_t, f_t^h)$, where $f_v$ and $f_t$ denote image and text backbone modules respectively, and $f_v^h$ and $f_t^h$ represent corresponding projection heads mapping embeddings to a shared $d$-dimensional space. Given an image $I$ and its associated long-form caption $T_{long}$, we first segment $T_{long}$ into $M$ sentence-level subcaptions $\{T_{sub}^i\}_{i=1}^M$. We then extract the image's global and local features using $f_v$ and project them using $f_v^h$:

$$[v_{cls}, \ v_{loc}] = f_v^h(f_v(I)) \in \mathbb{R}^{(P+1) \times d}, \tag{1}$$

where $v_{cls} \in \mathbb{R}^d$ denoted the global [CLS] embedding of an image and $v_{loc} \in \mathbb{R}^{P \times d}$ are $P$ patch local embeddings.

Similarly, text embeddings are obtained from the text encoder:

$$\left[t_{eot}^{long}, \ t_{loc}^{long}\right] = f_t^h(f_t(T_{long})) \in \mathbb{R}^{(N+1) \times d}, \qquad \left[\{t_{eot,i}^{sub}\}_{i=1}^M, \ _-\right] = f_t^h\big(f_t\big(\{T_{sub}^i\}_{i=1}^M\big)\big) \in \mathbb{R}^{M \times d}. \tag{2}$$

where $t_{eot}^{long} \in \mathbb{R}^d$ and $t_{loc}^{long} \in \mathbb{R}^{K \times d}$ denoted the global [EOT] and K local embeddings of long text, while $t_{eot}^{sub} = \{t_{eot,i}^{sub}\}_{i=1}^M \in \mathbb{R}^{M \times d}$ denoted the global embeddings of $M$ subcaptions.

During training, every image is paired with a short summary caption and a longer detailed caption. Modern long-text augmentation pipelines commonly expand raw summary captions or generate full descriptions with LVLMs. Typically, the first sentence $t_{eot,1}^{sub}$ (or $t_{eot}^{short}$) of such a generated caption serves as the summary. To exploit this hierarchical structure, we define the global objective as:

$$\mathcal{L}_{\text{global}} = \mathcal{L}_{\text{contrast}}^{\text{batch}}(v_{cls}, \ t_{eot}^{long}) + \lambda_{\text{short}} \mathcal{L}_{\text{contrast}}^{\text{batch}}(v_{cls}, \ t_{eot}^{short}) \tag{3}$$

where $\lambda_{\text{short}}$ is a hyperparameter, and the batch-level contrastive loss $\mathcal{L}_{\text{contrast}}^{\text{batch}}$ pulls matched image–caption pairs $(v_{cls},\ t_{eot}^{long})$ and $(v_{cls},\ t_{eot,1}^{sub})$ closer in the shared embedding space while pushing apart the mismatch pairs within the batch. This global objective therefore aligns each image with both its comprehensive and concise textual descriptions.

## 3.2 Fine-grained Cross-modal Alignment

**Local Token Calibration**   In dense and highly-aligned image-text pairs, redundancy and ambiguity frequently occur in both local image patches and local text tokens. On the image side, as shown in previous works (Fu et al., 2024; Bolya et al., 2023), a large number of local patches generated by vision transformers are either redundant or ambiguous, often corresponding to non-salient backgrounds, repeated structures or regions lacking clear semantic content. Similarly, token embeddings from lengthy captions can be repetitive or weakly informative, which dilutes the effectiveness of cross-modal alignment. To mitigate these issues, we adopt aggregation network (Zong et al., 2022), as adaptive calibration mechanism for both visual and textual local embeddings. Specifically, given an input sequence of $N$ tokens, each with dimension $d$, we denote the input as $X \in \mathbb{R}^{N \times d}$. The aggregated output $X' \in \mathbb{R}^{N' \times d}$ is computed as:

$$X' = \text{SoftMax}\Big(\frac{W_q\, \sigma(XW_k)^\top}{\tau}\Big)X,\tag{4}$$

where $W_k \in \mathbb{R}^{d \times d_k}$ and $W_q \in \mathbb{R}^{N' \times d_k}$ are learnable projection matrices ($d_k < d$), which $N'/N = 0.5$ by default, $\sigma(\cdot)$ is a non-linear activation GELU (Hendrycks & Gimpel, 2016), and $\tau$ is a learnable temperature parameter. We apply calibration modules independently to visual patches $v_{loc} \in \mathbb{R}^{P \times d}$ and local long caption tokens $t_{loc}^{long} \in \mathbb{R}^{K \times d}$, yielding refined patches $v' = \{\tilde{v}_i\}_{i=1}^{rP} \in \mathbb{R}^{rP \times d}$ and refined words $t' = \{\tilde{t}_i\}_{i=1}^{rK} \in \mathbb{R}^{rK \times d}$.

To further leverage these semantic tokens for fine-grained matching, we propose two complementary alignment strategies: *token reconstruction alignment* and *subcaption–aggregated patch alignment* that operate on top of them.

**Token Reconstruction Alignment**   To align semantic words with their corresponding visual patches, we use the reduced sequences $v'$ and $t'$ as queries in a bidirectional dot-product attention:

$$A_{v \rightarrow t} = \text{SoftMax}\Big(\frac{v'\,(t')^\top}{\sqrt{d}}\Big), \qquad\qquad A_{t \rightarrow v} = \text{SoftMax}\Big(\frac{t'\,(v')^\top}{\sqrt{d}}\Big)\tag{5}$$

These matrices select, for every image patch, the most relevant text token and vice-versa, yielding cross-modal reconstructions $V' \in \mathbb{R}^{(rP) \times d}$ and $T' \in \mathbb{R}^{(rK) \times d}$ :

$$V' = \{\tilde{V}_i\}_{i=1}^{rP} = (A_{v \rightarrow t}\, t'), \qquad\qquad T' = \{\tilde{T}_i\}_{i=1}^{rK} = (A_{t \rightarrow v}\, v').\tag{6}$$

We introduce a self-sample alignment objective that applies two contrastive terms, one for images and one for text, to make every refined token consistent with its cross-modal reconstruction. Specifically, we impose contrastive losses for each token within the same sample; therefore, no cross-sample negatives are needed. This considerably reduced computation and memory costs over aligning patch-words pairs across a batch:

$$\mathcal{L}_{\text{recon}}^{\text{image}}(v', V') = \frac{1}{rP}\sum_{i=1}^{rP}\mathcal{L}_{\text{contrast}}^{\text{sample}}(\tilde{v}_i,\ \tilde{V}_i), \qquad \mathcal{L}_{\text{recon}}^{\text{text}}(t', T') = \frac{1}{rK}\sum_{i=1}^{rK}\mathcal{L}_{\text{contrast}}^{\text{sample}}(\tilde{t}_i,\ \tilde{T}_i).\tag{7}$$

The final **W**ord-**P**atch **R**econstruction (WPR) objective is simply the sum:

$$\mathcal{L}_{\text{Word}}(v', t') = \mathcal{L}_{\text{recon}}^{\text{image}}(v', V') + \mathcal{L}_{\text{recon}}^{\text{text}}(t', T')\tag{8}$$

which enforces mutual, token-wise agreement across modalities.

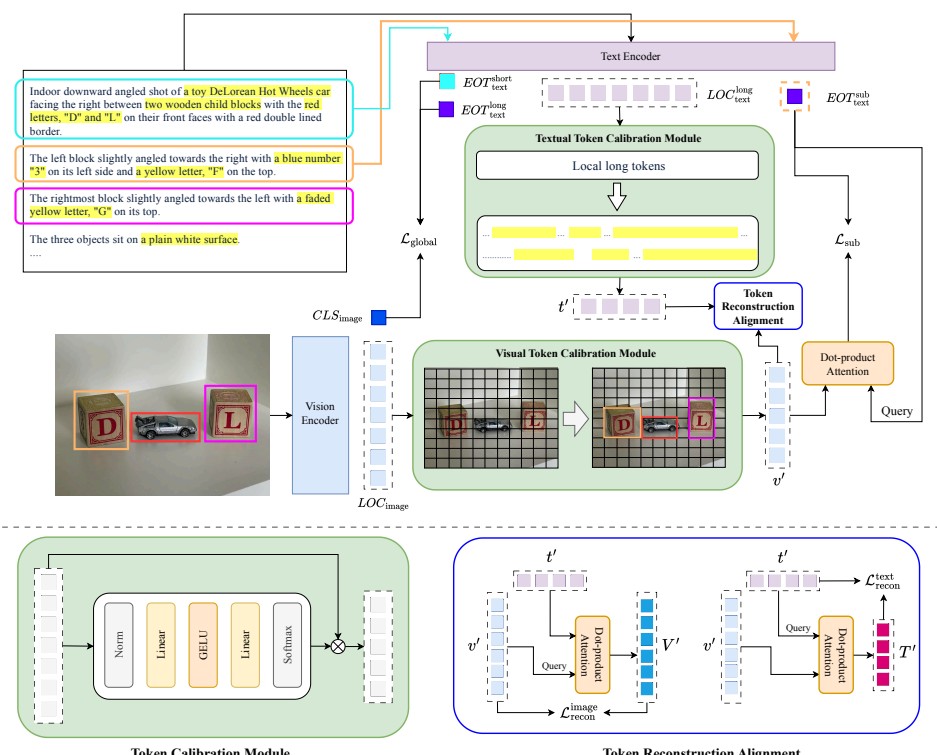

Figure 1: **Overview of MulCLIP**. An image encoder (ViT) produces a global image embedding $\text{CLS}_{img}$ and a sequence of local tokens $\text{LOC}_{img}$. The text encoder outputs local tokens $\text{LOC}_{text}$ and an end-of-text global embedding $\text{EOT}_{text}$ for multiple textual inputs, including long captions, summary captions, and other sub-captions. Independent calibration modules refine and shorten the local sequences of image and long text into $v'$ and $t'$. MulCLIP further exploits these semantic tokens through token reconstruction and the subcaption–aggregated patch mechanism

**Subcaption- Aggregated Patch Alignment** Descriptive captions from curated long-text image datasets typically consist of multiple sentence-level subcaptions, each can describe local image regions. To explicitly align these subcaptions with corresponding visual content, we obtain each subcaption embedding $t_{eot,i}^{\text{sub}} \in \mathbb{R}^{1 \times d}$ from Eq.2 and associate them with the aggregated visual representation from the refined local embeddings $v' \in \mathbb{R}^{rP \times d}$. Specifically, we use attention weights derived from dot-product similarity between each subcaption embedding and the visual patches:

$$\alpha^i = \text{SoftMax}\left(\frac{t_{eot,i}^{\text{sub}}(v')^\top}{\sqrt{d}}\right) \in \mathbb{R}^{1 \times rP}, \qquad \bar{v}^i = \alpha^i v' \in \mathbb{R}^d \qquad (9)$$

We then impose a **S**ubcaption-**A**ggregated **P**atch (SAP) objective that applies a contrastive loss between each subcaption embedding and its aggregated visual representation:

$$\mathcal{L}_{\text{Sub}}(v', t_{eot}^{sub}) = \frac{1}{M} \sum_{i=1}^{M} \mathcal{L}_{\text{contrast}}^{\text{batch}}(\bar{v}^i, t_{eot,i}^{sub}), \qquad (10)$$

**Overall Alignment Objective.** To enable robust and comprehensive vision–language alignment, we jointly optimize three complementary objectives:

$$\mathcal{L}_{total} = \mathcal{L}_{\text{global}} + \lambda_{\text{W}} \mathcal{L}_{\text{Word}}(v', t') + \lambda_{\text{S}} \mathcal{L}_{\text{Sub}}(v', t_{eot}^{sub}), \qquad (11)$$

where $\lambda_{\text{W}}, \lambda_{\text{S}}$ are weighting factors. We adopt a sigmoid-based contrastive loss (Zhai et al., 2023) as the main objective for all $\mathcal{L}_{\text{constrast}}$ terms.

| | Method | DOCCI | | | | | | DCI | | | | | | Avg |
|---|---|---|---|---|---|---|---|---|---|---|---|---|---|---|
| | | Text-to-Image | | | Image-to-Text | | | Text-to-Image | | | Image-to-Text | | | |
| | | R@1 | R@5 | R@25 | R@1 | R@5 | R@25 | R@1 | R@5 | R@25 | R@1 | R@5 | R@25 | |
| DOCCI FT / ViT-B/16 | FineLip | 70.94 | 92.98 | 96.82 | 71.50 | 93.24 | 97.38 | 58.58 | 78.64 | 85.09 | 59.88 | 80.39 | 86.34 | 80.98 |
| | GOAL | 79.47 | 96.65 | 99.69 | 79.43 | 96.14 | 99.61 | 64.13 | 82.69 | 92.95 | 65.88 | 83.44 | 92.95 | 86.09 |
| | **MulCLIP** | 82.2 | 97.12 | 99.78 | 80.26 | 96.88 | 99.67 | 69.08 | 85.99 | 93.44 | 67.13 | 84.24 | 94.75 | 87.55 |
| DOCCI FT / ViT-L/14 | FineLip | 74.70 | 94.24 | 97.32 | 75.44 | 94.60 | 97.72 | 62.88 | 81.69 | 87.14 | 63.68 | 83.44 | 88.29 | 83.43 |
| | GOAL | 84.37 | 99.55 | 99.76 | 82.57 | 97.37 | 99.82 | 68.93 | 85.74 | 93.95 | 68.43 | 85.99 | 93.90 | 88.37 |
| | **MulCLIP** | 86.73 | 98.10 | 99.84 | 84.80 | 97.88 | 99.84 | 72.93 | 88.00 | 94.94 | 72.03 | 86.64 | 95.65 | 89.78 |
| DCI FT / ViT-B/16 | Finelip | 65.50 | 89.30 | 94.92 | 66.32 | 90.72 | 95.24 | 60.38 | 80.39 | 86.79 | 63.58 | 82.94 | 88.39 | 80.37 |
| | GOAL | 71.22 | 92.39 | 98.90 | 71.18 | 92.88 | 98.88 | 72.64 | 89.89 | 95.95 | 72.84 | 90.50 | 96.60 | 86.99 |
| | **MulCLIP** | 73.78 | 93.86 | 99.04 | 71.75 | 92.96 | 99.26 | 75.13 | 89.44 | 95.90 | 72.00 | 89.24 | 96.34 | 87.55 |
| DCI FT / ViT-L/14 | FineLip | 68.84 | 90.92 | 95.36 | 71.54 | 92.56 | 96.58 | 66.03 | 84.49 | 89.29 | 65.58 | 85.19 | 90.40 | 83.07 |
| | GOAL | 79.04 | 95.78 | 99.55 | 79.16 | 95.96 | 99.61 | 76.89 | 91.05 | 96.55 | 76.59 | 91.20 | 96.55 | 89.83 |
| | **MulCLIP** | 81.04 | 96.33 | 99.54 | 78.35 | 95.31 | 99.54 | 78.83 | 91.39 | 96.79 | 76.83 | 92.09 | 97.34 | 90.28 |

Table 2: Long-text retrieval performance on DOCCI and DCI. Rows (DOCCI FT, DCI FT) indicate the dataset that methods was trained on, while columns (DOCCI, DCI) report evaluation performance. We highlight the models with best performance and second-best within each backbone, and gray shading indicates in-domain retrieval (diagonal blocks). MulCLIP improves the overall in-domain and out-of-domain performance on both datasets.

## 4 EXPERIMENTS

### 4.1 EXPERIMENTAL SETUP

**Datasets** We fine-tune MulCLIP on the training splits of both DOCCI and DCI (Onoe et al., 2024; Urbanek et al., 2024) and evaluate on the 3 test sets of DOCCI, DCI and Urban1K (Zhang et al., 2024) to measure both in-domain and out-of-domain fine-grained long-text retrieval performance comprehensively on Table 2 and Table 3 . The DOCCI dataset comprises 9,647 training examples and a test split totaling 5,100 samples (5,000 from the official test set plus 100 from the qualification set). To match this scale for DCI, whose original test partition contains only 100 examples, we follow (Choi et al., 2025) to randomly sampled 2,000 instances from its 7,805-sample training pool, yielding a comparable train–test ratio. To compare the short-text performance, we evaluate our model on the validation set of COCO2017 (Lin et al., 2015) and Flickr30k (Plummer et al., 2016).

**Training setting.** To validate our approach, we fine-tune two CLIP variants, ViT-B/16 and ViT-L/14, for 8 epochs, using a batch size of 16 for ViT-B/16 and 8 for ViT-L/14. Due to computational constraints, we use a smaller batch size for ViT-L/14 compared to the baseline (GOAL), which employs a batch size of 16. The total loss is a fixed weighted sum of global, detail, and token alignment terms $\lambda_{\text{short}} = 0.5, \quad \lambda_W = 1, \quad \lambda_S = 1$.

Training is performed on single NVIDIA A5000 GPU. We set the base backbone learning rate to $1 \times 10^{-5}$ and the refinement-module learning rate to $2 \times 10^{-4}$, so as to retain the pre-trained CLIP representations while encouraging the refinement layers to adapt to our long-caption datasets. A weight decay of 0.05 is applied to reduce overfitting, and we employ a linear warm-up over the first 200 iterations to stabilize the initial training phase.

**Test settings and state-of-the-art comparisons** We measure Text-to-Image (T2I) and Image-to-Text (I2T) retrieval performance using Recall@$k$. We compare MulCLIP against leading methods tailored for fine-grained, long-caption datasets, such as FineLIP and GOAL.

### 4.2 RESULTS

**In-domain Long Caption Retrieval.** On Tab. 2, MulCLIP establishes clear in-domain advantages on both DOCCI and DCI. On DOCCI, MulCLIP achieves the highest scores across all metrics and backbones, improving average R@1 over GOAL by nearly 2.5% and exceeding FineLIP by at least 10% in both T2I and I2T. On DCI, where description quality control is weaker than in DOCCI (Onoe et al., 2024), although GOAL benefits from its segment-filtering procedure, MulCLIP is able to achieve competitive performance with GOAL and continues to surpass FineLIP by a large margin.

**Zero-shot Long-Caption Cross-Modal Retrieval.** MulCLIP generalizes robustly across fine-grained long-text retrieval domains. When fine-tuned on DOCCI and tested on DCI (Tab. 2), it surpasses GOAL on T2I R@1 by about 5% with ViT-B/16 (69.1% vs. 64.1%) and about 4% with ViT-L/14 (72.9% vs. 68.9%). The trend persist in I2T where our model achieves an improvement of 3.1% with ViT-B/16 (84.0% vs. 81.9%) and about 2% with ViT-L/14 (88.3% vs. 86.3%). In the reverse setting (fine-tuned on DCI, evaluated on DOCCI), MulCLIP remains competitive with GOAL. A consistent performance gain is observed on Urban1k (Tab. 3), where MulCLIP achieves the highest recalls at nearly all thresholds, exceeding GOAL by at least 2% on both backbones.

| | Method | Urban1k | | | | | | Avg |
|---|---|---|---|---|---|---|---|---|
| | | Text-to-Image | | | Image-to-Text | | | |
| | | R@1 | R@5 | R@25 | R@1 | R@5 | R@25 | |
| ViT-B/16 | Zeroshot CLIP | 53.30 | 76.70 | 91.05 | 68.90 | 88.80 | 97.90 | 79.44 |
| | FineLIP DOCCI FT | 67.50 | 88.00 | 91.70 | 77.40 | 93.90 | 97.40 | 85.98 |
| | GOAL DOCCI FT | 73.20 | 92.70 | 98.30 | 81.90 | 95.80 | 99.40 | 90.22 |
| | **MulCLIP DOCCI FT** | 77.30 | 92.60 | 98.60 | 84.00 | 96.10 | 99.30 | 91.32 |
| | FineLIP DCI FT | 64.00 | 84.60 | 91.60 | 78.60 | 94.90 | 97.00 | 85.12 |
| | GOAL DCI FT | 77.20 | 93.70 | 98.60 | 82.90 | 96.80 | 99.40 | 91.43 |
| | **MulCLIP DCI FT** | 80.90 | 93.90 | 98.70 | 85.20 | 97.00 | 99.50 | 92.53 |
| ViT-L/14 | Zeroshot CLIP | 53.90 | 78.40 | 92.20 | 68.20 | 88.40 | 97.00 | 79.68 |
| | FineLIP DOCCI FT | 67.40 | 87.60 | 91.20 | 78.70 | 94.20 | 97.30 | 86.07 |
| | GOAL DOCCI FT | 83.00 | 95.40 | 99.70 | 86.30 | 96.50 | 99.40 | 93.38 |
| | **MulCLIP DOCCI FT** | 85.80 | 97.10 | 99.40 | 88.30 | 97.30 | 99.70 | 94.60 |
| | FineLIP DCI FT | 68.50 | 86.10 | 90.10 | 79.50 | 94.80 | 97.30 | 86.05 |
| | GOAL DCI FT | 84.50 | 96.40 | 99.50 | 89.80 | 97.80 | 99.60 | 94.60 |
| | **MulCLIP DCI FT** | 88.10 | 97.00 | 99.80 | 89.70 | 97.90 | 99.70 | 95.37 |

Table 3: Zero-shot cross-modal long-caption retrieval on Urban1k.

**Zero-shot Short Caption Retrieval.** After fine-tuning on long-caption data, MulCLIP still performs strongly on short-caption benchmarks. In many cases it improves over the pretrained CLIP baseline and tends to be stronger on T2I while staying competitive on I2T. For example, on Flickr30k, with ViT-L/14 trained on DCI, MulCLIP reaches I2T R@1 of 89.6% (vs. GOAL 88.1%, CLIP 86.7%); with ViT-B/16 trained on DOCCI, it attains T2I R@1 of 67.44% (vs. 66.92%, 63.20%). On COCO, it continues to lead T2I R@1 when trained on DCI for both backbones, and otherwise stays within roughly 1–2% of GOAL. For I2T, results are comparable, occasionally trailing GOAL by about 1–2%. Overall, MulCLIP preserves CLIP's short-caption strength while also delivering consistent improvements through long-caption fine-tuning.

| | Method | COCO | | | | Flickr30k | | | | Avg |
|---|---|---|---|---|---|---|---|---|---|---|
| | | Text-to-Image | | Image-to-Text | | Text-to-Image | | Image-to-Text | | |
| | | R@1 | R@5 | R@1 | R@5 | R@1 | R@5 | R@1 | R@5 | |
| ViT-B/16 | CLIP | 33.95 | 59.46 | 54.14 | 77.74 | 63.20 | 86.30 | 82.90 | 97.20 | 69.36 |
| | FineLIP DOCCI FT | 36.30 | 61.77 | 56.68 | 80.14 | 29.93 | 53.63 | 49.11 | 72.71 | 55.03 |
| | GOAL DOCCI FT | 37.28 | 62.96 | 56.84 | 80.20 | 66.92 | 88.56 | 83.20 | 96.70 | 71.58 |
| | **MulCLIP DOCCI FT** | 37.68 | 63.26 | 54.76 | 78.64 | 67.44 | 88.98 | 81.90 | 96.30 | 71.12 |
| | FineLIP DCI FT | 35.44 | 61.18 | 55.48 | 79.38 | 29.07 | 53.24 | 48.43 | 72.64 | 54.36 |
| | GOAL DCI FT | 37.20 | 63.17 | 55.82 | 79.10 | 66.12 | 88.42 | 82.70 | 96.60 | 71.14 |
| | **MulCLIP DCI FT** | 37.69 | 63.34 | 53.84 | 78.00 | 67.34 | 88.98 | 83.00 | 96.50 | 71.09 |
| ViT-L/14 | CLIP | 37.29 | 61.82 | 57.68 | 80.20 | 65.38 | 87.36 | 86.70 | 94.50 | 71.37 |
| | FineLIP DOCCI FT | 41.18 | 65.96 | 59.14 | 82.00 | 36.66 | 60.33 | 53.49 | 77.59 | 59.54 |
| | GOAL DOCCI FT | 44.22 | 69.19 | 62.82 | 84.04 | 73.88 | 92.22 | 89.80 | 98.60 | 76.85 |
| | **MulCLIP DOCCI FT** | 43.69 | 69.73 | 60.76 | 83.00 | 74.68 | 92.86 | 88.40 | 98.30 | 76.43 |
| | FineLIP DCI FT | 40.95 | 65.70 | 58.80 | 81.94 | 36.30 | 60.22 | 52.36 | 76.56 | 59.10 |
| | GOAL DCI FT | 43.90 | 68.60 | 61.12 | 83.30 | 72.88 | 91.68 | 88.10 | 98.10 | 75.96 |
| | **MulCLIP DCI FT** | 44.25 | 69.20 | 62.86 | 83.44 | 74.04 | 92.44 | 89.60 | 98.50 | 76.79 |

Table 4: Zero-shot short caption retrieval on COCO and Flickr30k. MulCLIP shows competitive performance, often matching or exceeding GOAL across different metrics and model backbones.

| Variant | Global | LC | WPR | SAP | Fine-tuning objective |
|---|---|---|---|---|---|
| Global only | ✓ | | | | $\mathcal{L}_{\text{total}} = \mathcal{L}_{\text{global}}$ |
| W/o LC & w/o SAP | ✓ | | ✓ | | $\mathcal{L}_{\text{total}} = \mathcal{L}_{\text{global}} + \mathcal{L}_{\text{word}}(v, t)$ |
| W/o SAP | ✓ | ✓ | ✓ | | $\mathcal{L}_{\text{total}} = \mathcal{L}_{\text{global}} + \mathcal{L}_{\text{word}}(v', t')$ |
| W/o WPR | ✓ | ✓ | | ✓ | $\mathcal{L}_{\text{total}} = \mathcal{L}_{\text{global}} + \mathcal{L}_{\text{Sub}}(v', t_{eot}^{sub})$ |
| **MulCLIP (ours)** | ✓ | ✓ | ✓ | ✓ | $\mathcal{L}_{\text{total}} = \mathcal{L}_{\text{global}} + \mathcal{L}_{\text{word}}(v', t') + \mathcal{L}_{\text{Sub}}(v', t_{eot}^{sub})$ |

Table 5: **Fine-tuning objectives for MulCLIP variants**. 'LC' refers to **L**ocal **C**alibration modules for both branches. 'WPR' refers to **W**ord-**P**atch **R**econstruction loss. 'SAP' refers to **S**ubcaption-**A**ggregated **P**atch loss.

## 5 ABLATION STUDY & ANALYSIS

We conduct extensive ablation studies to evaluate the contribution of each component in MulCLIP, using checkpoints fine-tuned on DOCCI and tested on long/short image–text retrieval. We further report the degradation in zero-shot classification performance on CIFAR(Krizhevsky et al., 2009) and ImageNet variants(Recht et al., 2019; Hendrycks et al., 2021).

### 5.1 CORE COMPONENT CONTRIBUTION.

To highlight the role of each component in MulCLIP, we consider the variants summarized in Tab. 5. To compare against an alternative late-interaction design, we also evaluate a "MulCLIP w CLIM", which keeps the full MulCLIP objective but replaces the word-patch reconstruction with the Cross-modal Late Interaction Module (CLIM) (Asokan et al., 2025; Yao et al., 2021) operating over the refined local textual and visual tokens. Additional ablations with alternative design choices are reported in the supplementary material.

**Impact on Long-text Understanding** As shown in Tab. 6, the "W/o LC & w/o SAP" configuration-which combines the WPR objective with global alignment—already yields substantial gains on Urban1k and DCI, most notably on the ViT-L/14 backbone, without degrading the in-domain performance of the "global only" setting. This demonstrates that token-level word embeddings improve robustness and transferability in long-text retrieval. Building on this, when we integrate local calibration, the semantic word–patch objective works in concert with global alignment ("W/o SAP" row), further boosting performance for both backbones. This suggests that redundancy in image patches and long-text tokens can hinder alignment, consistent with observation from prior study (Asokan et al., 2025). Finally, when we add the SAP alignment, we provide an additional layer of fine-grained grounding, allowing completed MulCLIP to achieve the best overall results across all metrics. Replacing MulCLIP's word–patch reconstruction with the CLIM design leads to clear underperformance relative to our proposed approach. We design a simple yet effective strategy to use the completed natural structures of long text in CLIP model fine-tuning.

| | Method | Urban-1k | | | | DCI | | | | DOCCI | | | | Avg |
|---|---|---|---|---|---|---|---|---|---|---|---|---|---|---|
| | | T⇒I | | I⇒T | | T⇒I | | I⇒T | | T⇒I | | I⇒T | | |
| | | R@1 | R@5 | R@1 | R@5 | R@1 | R@5 | R@1 | R@5 | R@1 | R@5 | R@1 | R@5 | |
| ViT-B/16 | Global only | 71.2 | 90.7 | 80.9 | 95.7 | 65.0 | 83.0 | 63.9 | 82.8 | 81.1 | 97.1 | 79.9 | 96.3 | 82.30 |
| | W/o LC & w/o SAP | 71.5 | 91.9 | 79.6 | 95.2 | 65.9 | 82.5 | 63.2 | 83.0 | 80.6 | 96.8 | 79.3 | 96.2 | 82.14 |
| | W/o SAP | 74.4 | 91.6 | 80.1 | 95.2 | 65.4 | 84.0 | 64.1 | 83.2 | 80.6 | 96.9 | 78.9 | 96.5 | 82.58 |
| | W/o WPR | 73.1 | 91.7 | 80.0 | 95.5 | 66.4 | 84.7 | 65.6 | 85.1 | 82.9 | 97.3 | 81.6 | 96.7 | 83.38 |
| | **MulCLIP (ours)** | 77.3 | 92.6 | 84.0 | 96.1 | 69.1 | 86.0 | 67.1 | 84.2 | 82.2 | 97.1 | 80.3 | 96.9 | 84.41 |
| | MulCLIP w CLIM | 68.3 | 87.8 | 78.8 | 93.4 | 64.0 | 82.2 | 62.8 | 81.7 | 78.5 | 95.8 | 77.2 | 95.2 | 80.48 |
| ViT-L/14 | Global only | 81.7 | 95.0 | 83.5 | 95.9 | 70.2 | 85.7 | 68.0 | 85.0 | 83.9 | 97.4 | 81.2 | 96.9 | 85.37 |
| | W/o LC & w/o SAP | 85.8 | 96.1 | 85.2 | 96.5 | 71.4 | 86.3 | 67.7 | 84.8 | 84.1 | 97.6 | 81.2 | 96.8 | 86.12 |
| | W/o SAP | 85.0 | 96.8 | 87.3 | 96.5 | 71.9 | 87.6 | 68.5 | 86.4 | 85.8 | 97.6 | 83.6 | 97.4 | 87.03 |
| | W/o WPR | 80.6 | 95.5 | 85.6 | 97.1 | 72.2 | 87.4 | 71.8 | 87.4 | 86.0 | 98.3 | 84.3 | 97.9 | 87.01 |
| | **MulCLIP (ours)** | 85.8 | 97.1 | 88.3 | 97.3 | 73.7 | 88.2 | 70.8 | 86.9 | 86.7 | 98.1 | 84.8 | 97.9 | 87.97 |
| | MulCLIP w CLIM | 82.7 | 95.5 | 84.7 | 95.3 | 71.6 | 87.2 | 70.4 | 87.0 | 84.5 | 97.8 | 83.4 | 97.4 | 86.46 |

Table 6: **Module ablations on long-text retrieval** over Urban-1k, DCI, and DOCCI. Using all three modules (LC, WPR, SAP) in MulCLIP yields the strongest performance among its variants.

| Method | Cifar | | ImageNet | | COCO | | | | Flickr | | | | Avg |
|---|---|---|---|---|---|---|---|---|---|---|---|---|---|
| | 10 | 100 | v2 | O | T⇒I | | I⇒T | | T⇒I | | I⇒T | | |
| | | | | | R@1 | R@5 | R@1 | R@5 | R@1 | R@5 | R@1 | R@5 | |
| Zeroshot CLIP | 90.80 | 67.30 | 61.90 | 42.20 | 37.29 | 61.82 | 57.68 | 80.20 | 63.20 | 86.30 | 82.90 | 97.20 | 69.07 |
| *ViT-B/16* Global only | 86.33 | 55.19 | 50.62 | 42.85 | 38.03 | 63.58 | 54.98 | 78.72 | 66.80 | 88.82 | 84.00 | 95.40 | 67.11 |
| W/o LC & w/o SAP | 85.48 | 58.39 | 51.49 | 42.60 | 38.02 | 64.09 | 55.40 | 79.36 | 67.92 | 88.94 | 84.80 | 96.30 | 67.73 |
| W/o SAP | 81.36 | 52.89 | 50.94 | 43.00 | 38.02 | 63.95 | 55.74 | 79.14 | 67.68 | 89.00 | 83.90 | 95.90 | 66.79 |
| W/o WPR | 84.98 | 55.07 | 51.82 | 41.65 | 37.12 | 62.71 | 54.88 | 78.00 | 65.74 | 88.30 | 83.00 | 96.50 | 66.65 |
| **MulCLIP (ours)** | 86.33 | 60.34 | 52.13 | 43.80 | 37.68 | 63.26 | 54.76 | 78.64 | 67.44 | 88.98 | 81.90 | 96.30 | 67.63 |
| MulCLIP w CLIM | 81.45 | 60.50 | 52.16 | 42.95 | 34.77 | 60.48 | 48.22 | 73.28 | 64.44 | 87.12 | 78.50 | 95.00 | 64.91 |
| Zeroshot CLIP | 95.50 | 76.80 | 69.90 | 31.90 | 37.29 | 61.82 | 57.68 | 80.20 | 65.38 | 87.36 | 86.70 | 94.50 | 70.42 |
| *ViT-L/14* Global only | 91.86 | 62.97 | 51.47 | 38.50 | 42.93 | 68.57 | 59.32 | 82.50 | 73.26 | 92.34 | 89.40 | 97.70 | 70.90 |
| W/o LC & w/o SAP | 90.31 | 64.34 | 54.29 | 37.35 | 38.17 | 64.09 | 55.40 | 79.36 | 74.24 | 92.76 | 88.70 | 98.30 | 69.78 |
| W/o SAP | 90.74 | 67.04 | 57.73 | 38.70 | 44.67 | 69.99 | 61.84 | 84.40 | 75.08 | 93.18 | 88.00 | 98.20 | 72.46 |
| W/o WPR | 91.71 | 67.79 | 56.95 | 36.25 | 43.22 | 68.78 | 60.92 | 83.28 | 74.30 | 92.36 | 88.10 | 98.50 | 71.85 |
| **MulCLIP (ours)** | 90.10 | 68.43 | 57.19 | 37.15 | 43.69 | 69.73 | 60.76 | 83.00 | 74.68 | 92.86 | 88.40 | 98.30 | 72.02 |
| MulCLIP w CLIM | 91.33 | 71.66 | 59.28 | 36.15 | 42.66 | 67.82 | 60.40 | 82.72 | 72.22 | 92.20 | 86.80 | 98.10 | 71.78 |

Table 7: **Module ablations on short-text understanding** across CIFAR-10/100 and ImageNet-v2/O classification (top-1 accuracy), and COCO/Flickr short-text retrieval.

**Impact on short-text understanding.** As shown in Tab. 7, the "W/o SAP" configuration, which includes global, local calibration and word–patch reconstruction, achieves the strongest short-text retrieval performance on COCO and Flickr for both backbones. Howevers, the full MulCLIP model and the "W/o WPR" variant, while improving ImageNet classification, slightly reduce retrieval performance on short-caption datasets. This trade-off may stem from SAP: introducing coherent sub-captions aligned with local visual regions helps longer descriptions but can act as noisy supervision once taken out of their full context. Overall, the complete MulCLIP improves the performance of pretrained CLIP on standard retrieval benchmarks, while show less degradation on zeroshot classification.

## 5.2 FINE-GRAINED ANALYSIS

### 5.2.1 FINE-GRAINED UNDERSTANDING ACROSS DIFFICULTY LEVELS

While our previous experiments primarily assess image-level retrieval, they mainly capture how well a model aligns global scene semantics with long or short descriptions. To explicitly probe local grounding, we further evaluate MulCLIP on the fine-grained FG-OVD benchmark, which is defined over localized regions rather than full images.

In FG-OVD, each region is annotated with one positive caption and a set of perturbed negatives created by replacing specific attribute words such as color, material, or

Table 8: **Fine-grained understanding on FG-OVD**. Accuracy (%) on the four difficulty subsets (hard, medium, easy, trivial) for different methods, all using a ViT-B/16 backbone fine-tuned on DOCCI.

| Method | hard | medium | easy | trivial | Avg |
|---|---|---|---|---|---|
| FineLIP | 18.17 | 38.68 | 41.96 | 73.79 | 43.15 |
| GOAL | 18.65 | 39.66 | 44.50 | 72.78 | 43.90 |
| **MulCLIP** | 19.24 | 40.73 | 47.27 | 68.63 | 43.97 |
| W/o SAP | 16.56 | 37.84 | 43.03 | 65.84 | 40.82 |
| W/o WPR | 17.38 | 38.51 | 45.42 | 68.41 | 42.43 |

spatial relations. These candidates are grouped into four difficulty levels—hard, medium, easy, and trivial—depending on how similar the negatives remain to the positive description, with the hardest cases differing by only one or two attributes. Following the standard protocol, we rank each region's true caption among its candidates. As shown in Tab. 8, MulCLIP consistently outperforms the other adaptation methods on the hard, medium, and easy splits, confirming that its multi-level alignment enhances sensitivity to subtle attribute changes.

### 5.2.2 QUALITATIVE LOCALIZATION RESULTS

Figure 2 compares ViT-B/16 attention maps of **GOAL**, the ablations (**W/o SAP**, **W/o WPR**), and our full **MulCLIP** model. MulCLIP consistently captures local details more precisely than any of the baselines. Both MulCLIP and its ablations can detect subtle cues such as the camouflaged long-tailed lizard on the rocks and black letters or the reflection of a car in mirrors. However, while the ablations

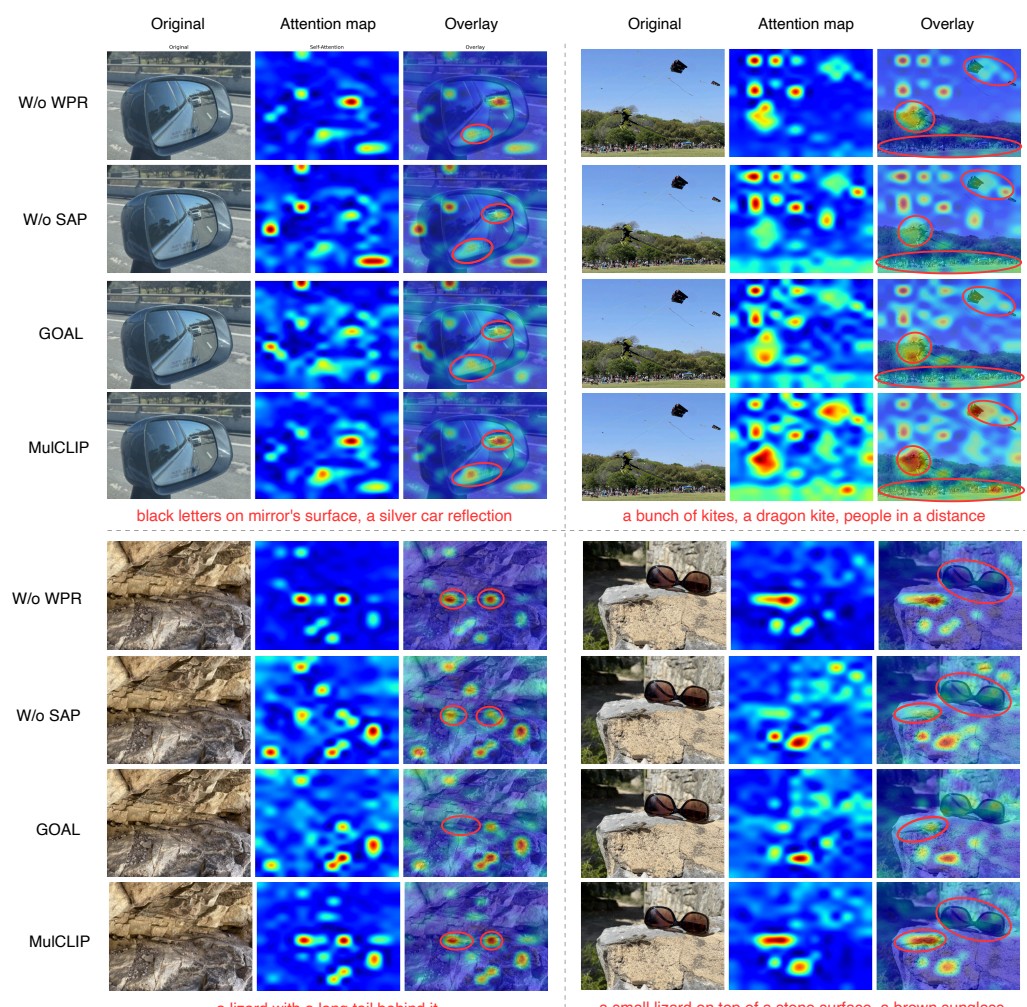

Figure 2: **Qualitative comparison of attention maps.** From left to right, we show: (1) the original image, (2) the attention heatmap, and (3) the overlay of the heatmap on the image. Across diverse scenes, MulCLIP produces sharper and more semantically aligned attention, successfully localizing fine-grained details that are often missed or diluted in baseline methods. Red circles highlight regions where MulCLIP demonstrates effective attention localization.

attend to these details, MulCLIP produces sharper and more semantically aligned activations; in contrast, "W/o SAP" and GOAL yield more diffuse responses, whereas "W/o WPR" produces less diffuse but more fragmented patterns that often miss broader contextual regions (i.e the eyeglass, people in a distance). Notably, GOAL completely misses the camouflaged lizard despite its use of SAM-based region proposals to support localization, revealing a blind spot compared to MulCLIP's self-learned alignment mechanism. These qualitative comparisons reinforce the quantitative results, indicating that MulCLIP effectively balances global comprehension with fine-grained localization while avoiding the drawbacks of external region-proposal modules.

# 6    CONCLUSION

We presented MulCLIP, a simple yet effective adaptation framework that brings multi-scale alignment to CLIP-style models without relying on region-proposal tools. Comprehensive experiments on long-caption retrieval and zero-shot transfer demonstrate that explicitly coupling global, sentence-level, and word-level objectives consistently improves both in-domain accuracy and cross-domain robustness. Ablation studies further show that each alignment branch plays a complementary role and that the full model provides a stronger fine-grained understanding.

## REPRODUCIBILITY STATEMENT

We aim to make our results straightforward to verify. Sections 3 and 4 document the implementation, model architectures, training/evaluation protocols, and all hyperparameters. To preserve double-blind review, the full source code and scripts will be released upon acceptance. During the rebuttal phase, if requested by reviewers or area chairs, we will provide an *anonymous* artifact bundle (e.g., source code, minimal pretrained checkpoints, configuration files, and step-by-step commands) via an anonymized URL compliant with the ICLR anonymity policy. All experiments use fixed random seeds; environment details are reported. Pretrained checkpoints and any preprocessed data will be shared subject to licensing constraints.

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

## A APPENDIX

### I. ZERO-SHOT CLASSIFICATION

Table 9 sums up the zero-shot results. With ViT-B/16, MulCLIP is consistently higher than GOAL on all datasets for both DOCCI and DCI fine-tuning. With ViT-L/14, the picture is mixed: under DOCCI fine-tuning, GOAL leads on CIFAR-10/100 and ImageNet-V2, while MulCLIP is stronger on ImageNet-O; under DCI fine-tuning, MulCLIP improves on ImageNet-O, ImageNet-V2, and CIFAR-10, with GOAL slightly ahead on CIFAR-100.

| | Method | Top-1 Accuracy (%) | | | | |
|---|---|---|---|---|---|---|
| | | CIFAR-100 | ImageNet-O | ImageNet-V2 | CIFAR-10 | Avg |
| ViT-B/16 | GOAL DOCCI FT | 55.41 | 42.15 | 49.85 | 84.95 | 58.09 |
| | MulCLIP DOCCI FT | 60.34 | 43.80 | 52.13 | 86.33 | 60.65 |
| | GOAL DCI FT | 57.70 | 40.85 | 53.19 | 86.16 | 59.48 |
| | MulCLIP DCI FT | 60.81 | 41.95 | 54.77 | 86.90 | 61.11 |
| ViT-L/14 | GOAL DOCCI FT | 69.61 | 33.90 | 63.25 | 93.70 | 65.12 |
| | MulCLIP DOCCI FT | 68.43 | 36.95 | 56.79 | 90.10 | 63.07 |
| | GOAL DCI FT | 73.03 | 32.50 | 61.17 | 92.07 | 64.69 |
| | MulCLIP DCI FT | 71.14 | 34.00 | 63.37 | 92.56 | 65.27 |

Table 9: Zeroshot top-1 accuracy classification performance on DOCCI and DCI checkpoints. We highlight the models with best performance.

## II. ABLATION OF MULCLIP WITH DIFFERENT CHOICES OF WORD-PATCH LATE INTERACTION.

**Ablation protocol.** Table 10 compares MulCLIP against three ablated variants that modify the word– patch objective:

(i) "*MulCLIP w/ Text-recon*" is the full framework but sets $\mathcal{L}_{\text{Word}} = \mathcal{L}_{\text{recon}}^{\text{text}}$;

(ii) "*MulCLIP w/ Image-recon*" is the full framework but sets $\mathcal{L}_{\text{Word}} = \mathcal{L}_{\text{recon}}^{\text{image}}$;

(iii) "*MulCLIP w/o Recon*" (naive approach) is the full framework but replaces token reconstruction with a batch-contrastive alignment between refined tokens and patches, which sets $\mathcal{L}_{\text{Word}} = \mathcal{L}_{\text{contrast}}^{\text{batch}}(v', t')$. Here $v'$ and $t'$ denote refined patch and token embeddings, respectively;

Across datasets and metrics, the full MulCLIP consistently delivers competitive performance, often matching or surpassing all baselines across backbones. When reconstruction is restricted to a single direction, the model remains effective on short captions, where one-to-one cues dominate. However, such one-sided objectives and naive approach reveal consistent shortcomings in cross-domain long-text transfer. By contrast, the full bidirectional scheme balances both perspectives and avoids collapsing into a single retrieval path, leading to more stable results under distribution shifts.

| Method | Urban-1K | | | | DCI | | | | DOCCI | | | | COCO | | | | Avg |
|---|---|---|---|---|---|---|---|---|---|---|---|---|---|---|---|---|---|
| | T⇒I | | I⇒T | | T⇒I | | I⇒T | | T⇒I | | I⇒T | | T⇒I | | I⇒T | | |
| | R@1 | R@5 | R@1 | R@5 | R@1 | R@5 | R@1 | R@5 | R@1 | R@5 | R@1 | R@5 | R@1 | R@5 | R@1 | R@5 | Avg |
| MulCLIP w/ Text-recon | 73.80 | 92.00 | 81.10 | 95.40 | 65.38 | 83.54 | 65.88 | 84.59 | 82.27 | 97.25 | 80.94 | 96.92 | 37.52 | 63.23 | 55.06 | 78.28 | 77.07 |
| MulCLIP w/ Image-recon | 73.20 | 91.60 | 80.80 | 94.90 | 67.08 | 84.89 | 65.73 | 84.79 | 82.71 | 97.45 | 80.78 | 96.92 | 37.31 | 63.28 | 54.52 | 78.16 | 77.13 |
| MulCLIP w/o Reconstruction | 73.90 | 92.30 | 80.60 | 95.60 | 66.88 | 83.84 | 66.68 | 85.04 | 82.73 | 97.00 | 80.96 | 96.90 | 37.19 | 62.78 | 54.26 | 78.14 | 77.18 |
| **MulCLIP (ours)** | 77.30 | 92.60 | 84.00 | 96.10 | 69.10 | 86.00 | 67.10 | 84.20 | 82.20 | 97.10 | 80.30 | 96.90 | 37.58 | 63.26 | 54.76 | 78.64 | 77.95 |
| MulCLIP w/ Text-recon | 86.20 | 96.50 | 87.60 | 97.00 | 73.74 | 87.74 | 71.24 | 87.84 | 86.47 | 98.22 | 84.33 | 97.78 | 43.90 | 69.61 | 61.30 | 83.66 | 82.07 |
| MulCLIP w/ Image-recon | 84.80 | 96.90 | 86.60 | 97.20 | 73.54 | 88.84 | 72.14 | 88.54 | 86.02 | 98.41 | 84.67 | 97.96 | 44.02 | 69.54 | 61.50 | 84.02 | 82.17 |
| MulCLIP w/o Reconstruction | 83.00 | 95.20 | 87.20 | 96.50 | 72.54 | 87.79 | 71.29 | 87.84 | 86.27 | 98.35 | 84.12 | 97.94 | 43.48 | 68.86 | 60.60 | 83.24 | 81.51 |
| **MulCLIP (ours)** | 85.80 | 97.10 | 88.30 | 97.30 | 73.70 | 88.20 | 70.80 | 86.90 | 86.70 | 98.10 | 84.80 | 97.90 | 43.69 | 69.73 | 60.76 | 83.00 | 82.05 |

Table 10: Ablation of MulCLIP with different word–patch late-interaction objectives. All rows use the checkpoint fine-tuned on DOCCI.

## III. EXTENDED RETRIEVAL QUALITATIVE RESULTS

Figures 7 and Table 16 illustrate a recurring limitation of **GOAL**: it often misses *small or low-contrast* details such as tiny numbers, faint text, background signs, or small logos. **MulCLIP** overcomes this through *multi-level alignment*, when we start from global fine-tuning and introduce raw word–patch alignment. It ensures that subtle cues, like route numbers, street-name plates, or curb textures, are preserved rather than averaged out. In practice, this leads to fewer sign mismatches, fewer counting errors, and more accurate grounding of in-image text. These qualitative improvements are consistent with the quantitative gains observed on urban retrieval benchmarks.

## IV. OPEN-VOCABULARY DETECTION EVALUATION (FG-OVD)

**Setup.** To further probe MulCLIP's fine-grained localization ability, we follow the open-vocabulary detection (FG-OVD) evaluation protocol of FG-CLIP (Xie et al., 2025). We plug different vision–language backbones into the official FG-CLIP detection pipeline, keeping the detector, training hyperparameters, and data splits fixed, with all three models fine-tuned on Docci. Using the same ViT-B/16 backbone, we re-evaluate MulCLIP, GOAL, and FineLIP on the four FG-OVD difficulty levels (*hard/medium/easy/trivial*).

| Method | Backbone | Hard | Medium | Easy | Trivial | Avg |
|---|---|---|---|---|---|---|
| FG-CLIP | ViT-B/16 | **46.10** | **66.60** | **68.70** | **83.40** | **66.20** |
| MulCLIP | ViT-B/16 | 19.24 | 40.73 | 47.27 | 68.63 | 43.97 |
| FineLIP | ViT-B/16 | 18.17 | 38.88 | 41.96 | 73.79 | 43.20 |
| GOAL | ViT-B/16 | 18.65 | 39.66 | 44.50 | 72.78 | 43.90 |

Table 11: **Open-vocabulary detection (FG-OVD).** Results under the official FG-CLIP pipeline with a shared ViT-B/16 backbone.

As expected, FG-CLIP is clearly best on FG-OVD, since it is trained with region-level supervision and a detection-oriented objective. In contrast, MulCLIP is only fine-tuned for long-/short-caption retrieval, without any box-level labels. Despite this, MulCLIP slightly outperforms GOAL and FineLIP on the hard, medium, and easy splits, and remains competitive on the trivial split (Table 11). This indicates that our multi-level alignment (Global + LC + WPR + SAP) transfers some fine-grained localization ability to an open-vocabulary detection setting, even though a dedicated OVD model like FG-CLIP still remains clearly stronger overall.

## V. SENSITIVITY TO LOCAL-LOSS WEIGHTS

**Setup.** To examine sensitivity to the local losses, we tie the two local weights and sweep $\lambda_{word} = \lambda_{sub} \in \{0.2, 0.6, 0.8, 1.0\}$ on the ViT-B/16 checkpoint fine-tuned on DOCCI. For each setting, we evaluate R@1 on long-text benchmarks (DOCCI, DCI, Urban1K) and short-text benchmarks (Flickr30K, COCO), as summarized in Table 12.

| $\lambda_{word} = \lambda_{sub}$ | Text-to-Image R@1 (%) | | | | | Image-to-Text R@1 (%) | | | | | Avg |
|---|---|---|---|---|---|---|---|---|---|---|---|
| | DOCCI | DCI | Urban1K | Flickr30K | COCO | DOCCI | DCI | Urban1K | Flickr30K | COCO | |
| 0.2 | 82.2 | 66.9 | 72.6 | 67.1 | 37.4 | 80.3 | 64.1 | 82.0 | **84.4** | **55.1** | 69.21 |
| 0.6 | **82.6** | 67.3 | 74.0 | 66.8 | 37.6 | **80.7** | 66.1 | 81.9 | 82.2 | 54.8 | 69.40 |
| 0.8 | 82.2 | 66.3 | 72.2 | 66.7 | 37.5 | **80.7** | 65.2 | 82.0 | 81.4 | 54.8 | 68.90 |
| 1.0 | 82.2 | **69.1** | **77.3** | **67.4** | **37.7** | 80.3 | **67.1** | **84.0** | 81.9 | 54.8 | **70.18** |

Table 12: **Ablation of tied local-loss weight** $\lambda_{word} = \lambda_{sub}$ **(ViT-B/16, DOCCI FT).** We report R@1 (%) for text-to-image (T⇒I) and image-to-text (I⇒T) retrieval on long-text (DOCCI, DCI, Urban1K) and short-text (Flickr30K, COCO) benchmarks.

When we vary $\lambda_{word} = \lambda_{sub}$ from 0.2 to 1.0, both long-text (DOCCI/DCI/Urban1K) and short-text (Flickr30K/COCO) R@1 scores change by at most about 1–2 points. In-domain performance on DOCCI is almost flat, while DCI and Urban1K show mild gains as $\lambda$ increases. Our default choice $\lambda = 1.0$ slightly favors long-text retrieval (especially on DCI and Urban1K) without noticeably degrading short-text performance. Overall, these results indicate that MulCLIP is robust with respect to the local-loss weights within a broad mid-range.

## VI. ROBUSTNESS TO NUMBER OF SUBCAPTIONS

**Setup.** We study how sensitive MulCLIP is to the number of sentence-level subcaptions. We fine-tune ViT-B/16 on DOCCI while varying the maximum number of sentences per caption from 5 to 20, and evaluate R@1 on long-text (DOCCI, DCI, Urban1K) and short-text (Flickr30K, COCO) retrieval. Subcaptions are defined at the sentence level using punctuation-based splitting.

| Max sentences | Text-to-Image R@1 (%) | | | | | Image-to-Text R@1 (%) | | | | | Avg |
|---|---|---|---|---|---|---|---|---|---|---|---|
| | DOCCI | DCI | Urban1K | Flickr30K | COCO | DOCCI | DCI | Urban1K | Flickr30K | COCO | |
| 5 | **82.9** | 66.4 | 76.2 | 65.6 | 37.1 | 81.1 | 65.5 | 81.2 | 81.5 | 54.2 | 69.17 |
| 10 | 82.3 | 65.8 | 74.8 | 66.3 | 37.3 | **81.4** | 66.7 | 82.1 | 81.9 | **54.5** | 69.31 |
| 15 (default) | 82.6 | 67.5 | 75.3 | 66.1 | 36.9 | 80.9 | 66.5 | 81.1 | **82.3** | 54.3 | 69.35 |
| 20 | 82.2 | **69.0** | **77.5** | **67.4** | **37.9** | 80.3 | **67.1** | **83.9** | 81.9 | 53.8 | **70.10** |

Table 13: **Effect of caption granularity (max sentences per caption).** R@1 (%) for text-to-image (T⇒I) and image-to-text (I⇒T) retrieval on long-text (DOCCI, DCI, Urban1K) and short-text (Flickr30K, COCO) benchmarks.

Across the range from 5 to 20 sentences, short-text R@1 on Flickr30K and COCO remains almost flat. DOCCI and DCI show small gains when increasing from very few sentences to around 10–20, after which performance saturates. Urban1K shows a mild upward trend, but improvements are incremental and never collapse.

Sentence-count histograms for DOCCI and DCI (Figures 5–6) show that most captions contain 3–10 sentences, with only a small fraction exceeding 20. Thus, our default cap of 15 sentences typically includes all available sentences without over-fragmenting the caption. Overall, MulCLIP benefits

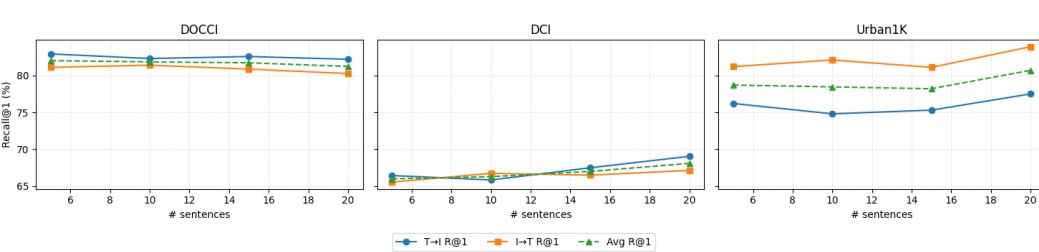

Figure 3: Effect of maximum number of sentences on long-text retrieval (DOCCI / DCI / Urban1K).

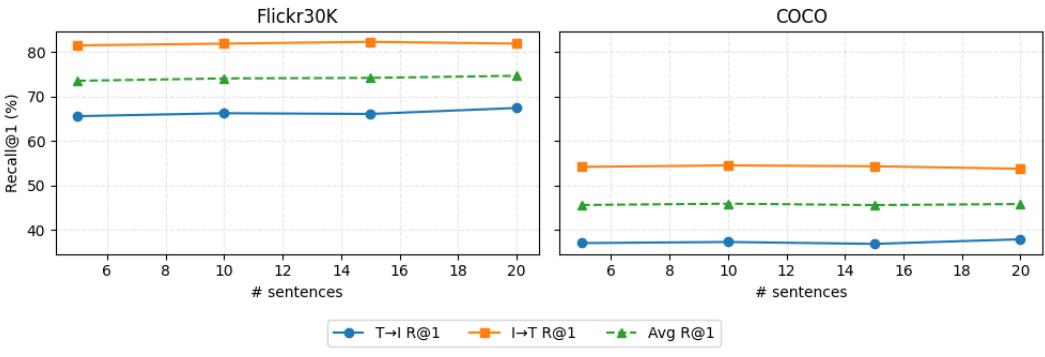

Figure 4: Effect of maximum number of sentences on short-text retrieval (Flickr30K / COCO).

from multiple sentence-level subcaptions but remains stable across a wide range of reasonable caps, indicating robustness to caption granularity.

## VII. FAIR COMPARISON WITH FINELIP AND ROLE OF THE GLOBAL LOSS

**Total loss and local modules.** For clarity, the full MulCLIP objective can be written as

$$\mathcal{L}_{\text{total}} = \mathcal{L}_{\text{global}}\big(v_{\text{cls}}, t_{\text{eot}}^{\text{long}}, t_{\text{eot}}^{\text{short}}\big) + \lambda_{\text{W}}\,\mathcal{L}_{\text{Word}}\big(v', t'\big) + \lambda_{\text{S}}\,\mathcal{L}_{\text{Sub}}\big(v', t_{\text{eot}}^{\text{sub}}\big), \tag{12}$$

where $v_{\text{cls}}$ is the global image embedding, $t_{\text{eot}}^{\text{long}}, t_{\text{eot}}^{\text{short}}$ are global text embeddings for long and short captions, and $v', t'$ are locally calibrated tokens used by the word- and subcaption-level objectives.

**Removing the global objective.** To isolate the contribution of our local modules, we train a variant that *removes* the global loss and keeps only local alignment:

$$\mathcal{L}_{\text{total}} = \mathcal{L}_{\text{Word}}(v', t') + \mathcal{L}_{\text{Sub}}(v', t_{\text{sub}}). \tag{13}$$

As shown in Table 14, this "No Global" model suffers a large drop on all three long-text benchmarks compared to full MulCLIP, with R@1 roughly halved in many cases. This confirms that local objectives alone are not sufficient for robust long-text understanding, and that they must work *together* with a strong global alignment term.

**Adding our global loss to FineLIP.** We next equip FineLIP with the *same* long/short global objective and 50% token compression as MulCLIP. Let

$$V = v' \oplus v_{\text{cls}}, \quad T = t' \oplus t_{\text{eot}} \tag{14}$$

be the concatenation of global and local tokens. The original FineLIP paper provides two runnable variants of its triplet-based CLIM/FILIP objective $R(\cdot)$: $R(V, T)$ and $R(v', t')$. We therefore define:

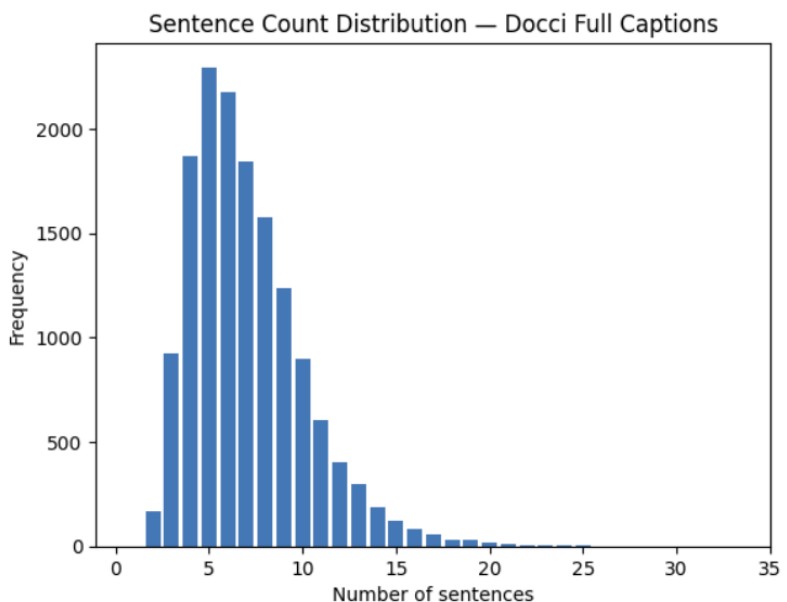

Figure 5: Sentence count distribution on DOCCI.

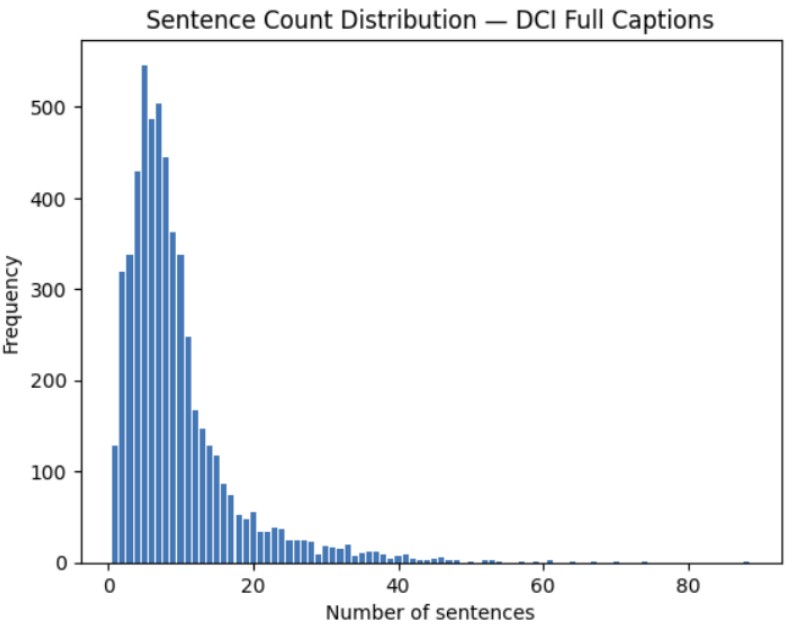

Figure 6: Sentence count distribution on DCI.

**FineLIP ver. 1 (+Global):**

$$\mathcal{L}_{\text{total}} = \mathcal{L}_{\text{global}}(v_{\text{cls}}, t_{\text{eot}}^{\text{long}}, t_{\text{eot}}^{\text{short}}) + R(V, T), \tag{15}$$

**FineLIP ver. 2 (+Global):**

$$\mathcal{L}_{\text{total}} = \mathcal{L}_{\text{global}}(v_{\text{cls}}, t_{\text{eot}}^{\text{long}}, t_{\text{eot}}^{\text{short}}) + R(v', t'). \tag{16}$$

For a fair comparison, we match these two FineLIP variants against our *W/o SAP* model

$$\mathcal{L}_{\text{total}} = \mathcal{L}_{\text{global}}(v_{\text{cls}}, t_{\text{eot}}^{\text{long}}, t_{\text{eot}}^{\text{short}}) + \mathcal{L}_{\text{Word}}(v', t'), \tag{17}$$

| Method | DCI | | DOCCI | | Urban1K | | Avg |
|---|---|---|---|---|---|---|---|
| | T⇒I | I⇒T | T⇒I | I⇒T | T⇒I | I⇒T | |
| MulCLIP (full) | **69.1** | **67.1** | **82.2** | **80.3** | **77.3** | **84.0** | **76.67** |
| No Global | 33.47 | 28.86 | 40.25 | 29.41 | 25.00 | 36.30 | 32.22 |

Table 14: **Effect of removing the global loss (ViT-B/16, DOCCI FT).** R@1 (%) for text-to-image (T⇒I) and image-to-text (I⇒T) retrieval.

so all methods share the same backbone, global loss, and token-compression ratio, differing only in how local interactions are modeled (FineLIP's CLIM/FILIP vs. our Word–Patch Reconstruction).

All three models are fine-tuned on DOCCI under the same protocol and evaluated on Urban1K, DCI, and DOCCI. Tables 15 summarize R@1 for both directions.

| | Method | Urban1K | | DCI | | DOCCI | | Avg |
|---|---|---|---|---|---|---|---|---|
| | | T⇒I | I⇒T | T⇒I | I⇒T | T⇒I | I⇒T | |
| ViT-B/16 | FineLIP (ver. 1, +Global) | 64.9 | 71.4 | 56.4 | 44.8 | 65.6 | 60.0 | 60.5 |
| | FineLIP (ver. 2, +Global) | 64.9 | 74.4 | 56.2 | 44.7 | 65.5 | 59.7 | 60.9 |
| | **Ours — W/o SAP** | **74.4** | **80.1** | **65.4** | **64.1** | **80.6** | **78.9** | **73.9** |
| ViT-L/14 | FineLIP (ver. 1, +Global) | 68.6 | 73.2 | 59.6 | 43.3 | 72.9 | 67.0 | 64.1 |
| | FineLIP (ver. 2, +Global) | 65.5 | 73.9 | 57.7 | 48.0 | 72.4 | 67.0 | 64.1 |
| | **Ours — W/o SAP** | **85.0** | **87.3** | **71.9** | **68.5** | **85.8** | **83.6** | **80.4** |

Table 15: **Fair comparison between FineLIP+Global and our Global+LC+WPR (W/o SAP).** All models share the same backbone, global loss, token-compression ratio, and DOCCI fine-tuning protocol. Best score per backbone is highlighted.

Under a fully matched setup (same backbone, global loss, token compression, data, and optimization), both FineLIP+Global variants remain consistently below our Global+LC+WPR (W/o SAP) model on all three long-text benchmarks, in both directions and for both ViT-B/16 and ViT-L/14. Since the only difference is how local tokens are used, this indicates that our Local Calibration and Word–Patch Reconstruction modules exploit compressed local tokens more effectively than FineLIP's CLIM/FILIP interaction.

The comparison with the "No Global" variant further highlights the complementarity of components: the global objective is essential for long-text robustness, while LC+WPR provide the additional fine-grained gains on top. In the main ablations, adding SAP on top of Global+LC+WPR then yields further, stable improvements, suggesting that subcaption–patch alignment is complementary rather than the sole driver of MulCLIP's benefits.

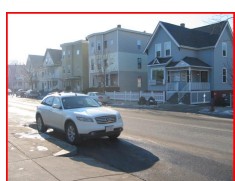 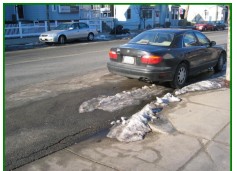

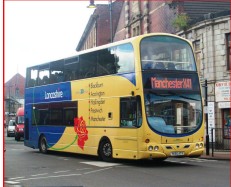 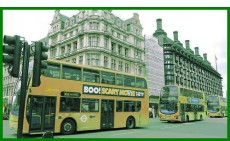

**Retrieved by GOAL**   **Retrieved by MulCLIP**       **Retrieved by GOAL**       **Retrieved by MulCLIP**

**Query:** *A* ==dark-colored sedan== *is parked askew on the side of a street, half on the asphalt road and half on the concrete sidewalk. Patches of melting snow are present, indicating recent snowfall or wintry conditions. In the background, a white sedan is parked correctly on the opposite side of the street. Residential buildings with white fences, bare deciduous trees, and other parked cars line the street. There's a general sense of a suburban or residential neighborhood on a clear day with sunlight casting shadows on the ground.* ==Visible tire tracks through the snow== *suggest recent vehicle movement.*

**Query:** *The image displays a vibrant urban scene with* ==two== *modern double-decker buses on a road, presumably in a city in the United Kingdom. The bus in the foreground is painted in a bright yellow color with bold advertising on its side, while* ==the bus in the background is also yellow== *with visible route information. Traffic lights appear on the left, indicating a crosswalk or intersection. European-style architecture is prominent, with elaborate stone buildings adorned with numerous windows and ornamental details. The sky is overcast with hints of* ==blue peeking through the clouds== *, suggesting a typical cloudy day. The greenery of trees is also visible, adding a touch of nature to the urban environment.*

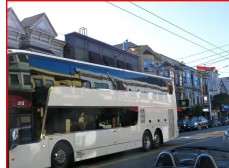 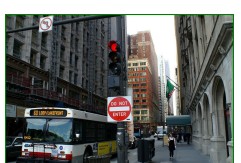

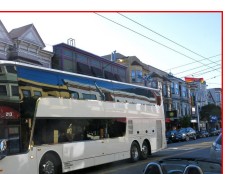 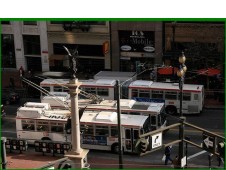

**Retrieved by GOAL**   **Retrieved by MulCLIP**       **Retrieved by GOAL**       **Retrieved by MulCLIP**

**Query:** *The image depicts a bustling city street scene with clear skies above. In the foreground, a white bus with a digital sign that reads* =='60 LOOP/LAKEFRONT'== *stops near a* ==sidewalk, marked 'K412'== *.* ==A red traffic light== *hangs above, while a* =='DO NOT ENTER' sign== *is prominently displayed on a post below. The architecture includes tall, ornamented stone buildings indicative of early 20th-century design, with one building featuring a scaffolding structure along its facade. A* ==pedestrian== *crosses the street, another walks on the sidewalk, and a few flags, including a green, white, and red one, are visible hanging from a building. The urban environment suggests a downtown district, possibly in a large metropolitan city.*

**Query:** *The image captures a busy urban street scene with* ==two== *white articulated trolleybuses,* ==featuring blue and red stripes== *, connected to overhead wires. Above the buses, a* ==streetlight== *with a dual-globe design is visible. In the foreground,* ==a pole topped with a flying eagle== *statue anchors the composition. Behind the buses, several red and white cars are parked. The backdrop is lined with multistory buildings hosting various stores with visible signage.* ==Pedestrians== *can be seen walking along the sidewalks, and traffic lights are located at the street's intersection. The photo, taken from an elevated angle during daylight, shows the street intersecting leftward, with designated lanes for different directions.*

**Legend:** ▊ Correct retrieval    ▊ Incorrect retrieval    ==Highlighted text== indicates visual details missing from GOAL's retrieved image but correctly matched by MulCLIP

Figure 7: Qualitative comparison of text-to-image retrieval between GOAL and MulCLIP. Each pair shows retrieved images from both models for the same query. Colored borders indicate correctness (green: correct; red: incorrect). Yellow highlights denote visual details missing from GOAL's retrieved image but correctly matched by MulCLIP.

| Image query | MulCLIP | GOAL |
|---|---|---|
| 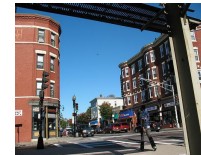 | This image depicts a vibrant urban street corner on a clear day with blue skies. A person in a blue checkered shirt and casual pants crosses the road at a pedestrian crosswalk, heading towards a series of red brick buildings with rounded and straight corners. The architecture suggests a charming, historic neighborhood with storefronts on the ground level, including one with a blue awning. There's a green streetlight visible and a black lamp post, adding to the quaint ambiance. Vehicles are stopped at the intersection, and the scene includes an overhanging metal structure that could be part of a bus stop. The overall atmosphere is that of a peaceful, sunny day in a bustling city neighborhood. | The image depicts an urban street scene during daytime. In the foreground, two individuals with their backs to the camera are walking, one with a long ponytail and a white shirt, the other with a patterned blouse, and a red backpack. A silver car is visible on the left side of the road, which is marked with multiple round blue traffic signs, indicating no waiting or no stopping restrictions. Across the street, there's a red-bricked five-story building with white stone trimmings and arched windows on the ground floor. The windows on the upper floors are regularly spaced, and the uppermost story appears to be an attic with smaller windows. The sky is clear, suggesting favorable weather conditions. |
| 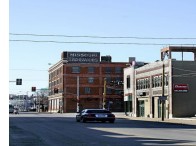 | The image shows an urban street scene under a clear blue sky. In the foreground, a black car is driving down the road, which is lined with electrical wires above. The architecture is a mix of multi-story brick buildings with visible signs of wear and faded paint, suggesting a historical urban area. The buildings vary in height, with some having flat facades and others featuring recessed windows and ornate detailing. The street appears to be relatively empty, with no pedestrians visible and minimal vehicular traffic. The structures' color palette is predominantly red brick, with accents of tan and white on the secondary building elements. | The image captures a daytime scene on a city street named "Main Street," indicated by a street sign hanging above. Vehicles, including a red sedan in the foreground, are parked along one side of the street, while others, including white vans, are visible in motion. Pedestrians are present on the sidewalks, some standing and others seated beside buildings; a group congregates near an American flag. Utility poles, traffic signals, and signs, including one indicating a "Drug-Free School Zone," dot the streetscape. Overhead, a concrete overpass spans the thoroughfare. The sky is slightly overcast, casting even lighting across the urban environment. |
| 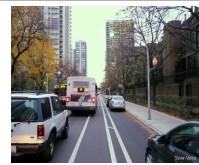 | This image captures an urban street scene with tall residential buildings lining one side and leafy trees displaying autumn colors. The scene includes a city bus in the center of the frame, showing the number 36 on its indicator, and various other vehicles such as cars and SUVs. The road features a dedicated bike lane on the right, demarcated by white lines and identified by painted bicycle symbols. The overcast sky and the presence of a streetlight that is turned on suggest that this is either early morning or late afternoon. The photo appears to be taken from the perspective of a pedestrian or cyclist at street level, focused on capturing the flow of urban traffic. | This image depicts an overcast day on an urban street lined with tall, modern office buildings. A blue public bus marked with the number 421 is at the forefront on the road, while a red bus can be seen farther down the street. There is a white car on the left and traffic lights are visible overhead, with a red light illuminated. The road has multiple lanes and a pedestrian zebra crossing in the foreground. There's also a traffic sign indicating no left turn for motorcycles. Leafless trees suggest it may be winter or early spring. The overall scene appears to be calm with moderate traffic. |
| 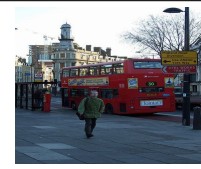 | This image captures a British urban scene, highlighted by a classic red double-decker bus on the right, displaying route number 30. The bus has yellow text and advertisements printed on its sides. On the left, a pedestrian wearing a green jacket and carrying a bag seems to be briskly walking on the sidewalk. There's a yellow street sign indicating a diversion ahead. In the background, an ornate building towers with a clock at its apex under a clear blue sky. The street is flanked by various other buildings, likely a mix of residential and commercial structures, typical of a UK cityscape. | This image captures a bustling urban scene, likely in London, with a red double-decker bus dominating the foreground, bearing the signage 'Arriva' and a route number 176 to Penge via Elephant & Castle and Forest Hill. A person at a pedestrian crossing is using a push-button signal post, while others wait by a bus stop shelter where someone points upwards. To the right, a classic red telephone box is in use by an individual. In the background, neoclassical architecture suggests a historical district, with a dome-topped building visible in the distance. The street is lined with cars and traditional black iron fencing, contributing to a distinctly British urban landscape. |

Table 16: Qualitative comparison of image-text retrieval results between MulCLIP (middle column) and GOAL (right column). Borders are embedded to indicate correctness (green: correct; red: incorrect). **Yellow highlights** denote visual details missing from GOAL's retrieved image but correctly matched by MulCLIP.

