# OpenReview forum: "MulCLIP: A Multi-level Alignment Framework for Enhancing Fine-grained Long-context CLIP"
_ICLR.cc/2026/Conference — Submitted to ICLR 2026_

### Official Review · Reviewer_ksab · 2025-10-28

**Soundness:** 2
**Presentation:** 2
**Contribution:** 2
**Rating:** 4
**Confidence:** 3

**Summary:**

The paper introduces MulCLIP, a novel end-to-end multi-level alignment framework that bridges short and long text structures with both global and regional image information, enabling fine-grained vision-language understanding. It proposes two key components — token reconstruction alignment and subcaption-aggregated patch alignment — to achieve three-level alignment. Experimental results demonstrate that MulCLIP consistently outperforms baseline models across multiple benchmarks.

**Strengths:**

The paper proposed two novel alignment strategies that enable multi-scale alignment, allowing CLIP to capture fine-grained textual and visual information. MulCLIP demonstrates gains over baselines across extensive benchmarks and multiple model scales. The paper provides detailed ablation studies, clearly evaluating the impact of each component.

**Weaknesses:**

1. The proposed components are largely adaptations of existing mechanisms, with the main contribution being their combination into a unified framework.
2. The performance depends on caption quality, which may be sensitive to noisy or weak textual annotations, and robustness under such conditions is not evaluated.
3. Experiments focus primarily on retrieval benchmarks, with no evaluation on classification tasks (e.g., ImageNet or DataComp).

**Questions:**

1. How does MulCLIP perform on classification tasks such as ImageNet or DataComp?
2. Since token reconstruction alignment aims to reduce local redundancy, could it inadvertently remove semantically important patches or tokens?
3. How sensitive is MulCLIP to the number or granularity of subcaptions, and how does this affect overall performance?

---

> ### Author Response · Authors · 2025-11-21
> **Initial response 1**
>
> **Weakness 1**
> > “The proposed components are largely adaptations of existing mechanisms, with the main contribution being their combination into a unified framework.”
>
> **Response.** We appreciate this concern, but our local objectives are not simple reuses of FILIP/CLIM or GOAL. Our Word–Patch Reconstruction (WPR) loss works by reconstructing image and text tokens within each image–caption pair and then applying a contrastive loss on the reconstructed vs. original tokens. This self-supervised, pair-wise reconstruction (combined with LC token calibration) is different from late-interaction losses that match tokens across samples, and our ablation “MulCLIP w CLIM” under the same backbone and training setup consistently underperforms MulCLIP, indicating that WPR adds real value beyond the shared framework.
>
> **Subcaption-level alignment**. The SAP loss is also not a direct copy of existing region-alignment methods: it operates at the sentence (subcaption) level, aligning each sentence with an aggregated set of refined patches without relying on any external region-proposal tools. In contrast, prior works such as GOAL first invoke external detectors to extract candidate regions and then align those regions with short text, rather than modeling the full multi-sentence document structure jointly with local vision tokens. In our experiments, adding SAP on top of Global + LC + WPR yields consistent gains on long-text benchmarks, especially out-of-domain, showing that each MulCLIP component contributes beyond merely combining existing ideas.

---

> ### Author Response · Authors · 2025-11-21
> **Initial response 2**
>
> **Weakness 2 and Question 2**
> > “The performance depends on caption quality, which may be sensitive to noisy or weak textual annotations, and robustness under such conditions is not evaluated.”
> > "Since token reconstruction alignment aims to reduce local redundancy, could it inadvertently remove semantically important patches or tokens?"
>
> **Effect of token reconstruction on GOAL-filtered captions**. We thank the reviewer for giving us the opportunity to expand our experimental results. To test whether token reconstruction might discard important content, we train MulCLIP on GOAL-filtered DOCCI/DCI captions, which already focus on salient regions and contain little redundancy to “compress away”. For ViT-B/16, long-text retrieval on both DOCCI and DCI, under both in-domain and cross-domain evaluation, remains essentially the same as when training on the original captions, with R@1 differences staying within about 1–1.5 points and sometimes slightly favoring GOAL-filtered training. This pattern is consistent with a soft redundancy reduction mechanism: the reconstruction loss encourages redundant tokens and patches to share information, but does not aggressively prune unique, semantically important tokens, as evidenced by the stable performance even when almost every token in the caption is important.
>
> **ViT-B/16 — DOCCI fine-tuning**
>
> | Training Data | Eval Data | T→I R@1 | I→T R@1 |
> | :--- | :--- | ---: | ---: |
> | Original DOCCI | DOCCI | 82.20 | 80.26 |
> | GOAL-filtered | DOCCI | **82.57** | **80.35** |
> | Original DOCCI | DCI | **69.08** | **67.13** |
> | GOAL-filtered | DCI | 67.58 | 65.93 |
>
> **ViT-B/16 — DCI fine-tuning**
>
> | Training Data | Eval Data | T→I R@1 | I→T R@1 |
> | :--- | :--- | ---: | ---: |
> | Original DCI | DOCCI | **73.78** | **71.75** |
> | GOAL-filtered | DOCCI | 72.84 | 70.98 |
> | Original DCI | DCI | **75.13** | 72.00 |
> | GOAL-filtered | DCI | 74.84 | **72.19** |
>
> Beyond these results, we also provide attention-map ablations and experiments at different token compression ratios, as well as an additional study where we selectively remove the visual or textual LC modules (see our response to reviewer [**WQ3m**] and [**k6Eh**]), all of which support the conclusion that token reconstruction preserves important content while reducing redundancy.

---

> ### Author Response · Authors · 2025-11-21
> **Initial response 3**
>
> **Question 1 and Weakness 3**
> > "How does MulCLIP perform on classification tasks such as ImageNet or DataComp?"
>
> We do report zero-shot classification results in the appendix. We evaluate MulCLIP and GOAL checkpoints (ViT-B/16 and ViT-L/14) fine-tuned on DOCCI or DCI on four standard classification benchmarks: CIFAR-100, ImageNet-O, ImageNet-V2, and CIFAR-10.
>
> | Backbone | FT data | Method | CIFAR-100 | ImageNet-O | ImageNet-V2 | CIFAR-10 |
> | :--- | :--- | :--- | ---: | ---: | ---: | ---: |
> | ViT-B/16 | DOCCI | GOAL | 55.41 | 42.15 | 49.85 | 84.95 |
> | | | MulCLIP | **60.34** | **43.80** | **52.13** | **86.33** |
> | | DCI | GOAL | 57.70 | 40.85 | 53.19 | 86.16 |
> | | | MulCLIP | **60.81** | **41.95** | **54.77** | **86.90** |
> | ViT-L/14 | DOCCI | GOAL | **69.61** | 33.90 | **63.25** | **93.70** |
> | | | MulCLIP | 68.43 | **36.95** | 56.79 | 90.10 |
> | | DCI | GOAL | **73.03** | 32.50 | 61.17 | 92.07 |
> | | | MulCLIP | 71.14 | **34.00** | **63.37** | **92.56** |
>
> **Zero-shot classification after long-text adaptation**.
> For ViT-B/16, MulCLIP consistently outperforms GOAL after both DOCCI and DCI fine-tuning across all four classification benchmarks, with especially clear gains on CIFAR-100 and robustness-focused ImageNet-O. For ViT-L/14, the results are overall comparable: GOAL is slightly better on some in-distribution datasets, while MulCLIP is stronger on robustness benchmarks such as ImageNet-O and improves ImageNet-V2 under DCI fine-tuning, with only small drops elsewhere. These trends indicate that our multi-level alignment does not trade retrieval gains for weaker classification, but generally preserves or improves zero-shot accuracy, particularly for the base backbone. We did not extend this evaluation to large-scale DataComp due to compute constraints, but the current results already show that MulCLIP remains a strong zero-shot classifier after long-text adaptation.

---

> ### Author Response · Authors · 2025-11-21
> **Initial response 4**
>
> **Question 3**
> > "How sensitive is MulCLIP to the number or granularity of subcaptions, and how does this affect overall performance?"
>
> **Robustness to number of subcaptions.** We directly test how sensitive MulCLIP is to the number of subcaptions by fine-tuning ViT-B/16 on DOCCI while varying the maximum sentences per caption from 5 to 20, and then evaluating on both short-text (Flickr30K, COCO) and long-text (DOCCI, DCI, Urban1K) retrieval. Across this range, the R@1 curves on Flickr/COCO are almost flat, and on DOCCI/DCI we see a small improvement when moving from very few sentences to around 10–15, after which performance quickly saturates. Urban1K shows a mild upward trend with more sentences, but the gains are incremental rather than dramatic, and performance never collapses.
>
> | Max sentences | DOCCI T→I | DCI T→I | Urban1K T→I | Flickr30K T→I | COCO T→I | DOCCI I→T | DCI I→T | Urban1K I→T | Flickr30K I→T | COCO I→T |
> | :--- | ---: | ---: | ---: | ---: | ---: | ---: | ---: | ---: | ---: | ---: |
> | 5 | **82.9** | 66.4 | 76.2 | 65.6 | 37.1 | 81.1 | 65.5 | 81.2 | 81.5 | 54.2 |
> | 10 | 82.3 | 65.8 | 74.8 | 66.3 | 37.3 | **81.4** | 66.7 | 82.1 | 81.9 | **54.5** |
> | 15 (default) | 82.6 | 67.5 | 75.3 | 66.1 | 36.9 | 80.9 | 66.5 | 81.1 | **82.3** | 54.3 |
> | 20 | 82.2 | **69.0** | **77.5** | **67.4** | **37.9** | 80.3 | **67.1** | **83.9** | 81.9 | 53.8 |
>
> Table: Effect of caption granularity (maximum number of sentences per caption) on R@1 (%) for text-to-image (T→I) and image-to-text (I→T) retrieval. Subcaptions are defined at the sentence level (punctuation-based splitting). Performance is smooth across a wide range of sentence caps, with 15 sentences (our default) behaving similarly to nearby settings.
>
> *(Figures showing the effect of number of sentences on R@1: [Long text figure link](https://anonymous.4open.science/api/repo/mulclip_images-41C6/file/long.png?v=b2f0b455) and [Short text figure link](https://anonymous.4open.science/api/repo/mulclip_images-41C6/file/short.png?v=b12fe250))*
>
> Caption granularity and sentence statistics. Histograms of DOCCI and DCI show that most captions contain roughly 3–10 sentences, with only a small fraction having more than 20. In practice, this means our default choice (15 sentences) typically uses all available sentences instead of over-fragmenting the caption, and increasing the cap beyond that brings diminishing returns, consistent with the nearly flat curves. Crucially, subcaptions are defined at the sentence level (using punctuation-based rules) rather than by arbitrarily chopping text into short fragments, which keeps each subcaption semantically coherent. Taken together, these results indicate that MulCLIP is not overly sensitive to the exact number or granularity of subcaptions: it benefits from having multiple sentences, but behaves smoothly across a reasonable range of settings.
>
> *(Figures showing the sentence count distribution: [DOCCI distribution figure link](https://anonymous.4open.science/api/repo/mulclip_images-41C6/file/docci_dis.png?v=031593c9) and [DCI distribution figure link](https://anonymous.4open.science/api/repo/mulclip_images-41C6/file/dci_dis.png?v=485da0c6))*

---

> ### Author Response · Authors · 2025-11-26
> **Follow-up on rebuttal for Submission10277**
>
> Dear Reviewer,
>
> We are writing to kindly draw your attention to our submitted rebuttal. We have worked diligently to address all your valuable comments and have added the new experimental results you requested.
>
> We genuinely believe these updates align well with your vision for improving the paper. We respectfully ask that you consider re-evaluating the rating to reflect these substantial improvements.
>
> Thank you for your time and support.

---

### Official Review · Reviewer_gZLF · 2025-10-30

**Soundness:** 2
**Presentation:** 3
**Contribution:** 3
**Rating:** 4
**Confidence:** 4

**Summary:**

This paper introduces MulCLIP, a multi-level alignment framework that enhances fine-grained long-context understanding for vision-language models while preserving short-text performance. It achieves this through global alignment of images with long/summary captions, token reconstruction alignment between words and image patches, and subcaption-aggregated patch alignment, eliminating the need for external region-proposal tools. Experimental results show MulCLIP outperforms baselines in both long- and short-text retrieval tasks.

**Strengths:**

1. The paper is well-written and easy to follow.
2. The proposed token reconstruction alignment and subcaption-aggregated patch alignment strategies are interesting and innovative.
3. Experimental results reflect the effectiveness of the proposed method to some extent.

**Weaknesses:**

1. The claim in Lines 124-126 of the paper is inappropriate. In fact, MulCLIP does not achieve finer granularity than FG-CLIP, which uses region-proposal assistance; the two methods merely differ in their approaches to fine-grained alignment.
2. The Subcaption-Aggregated Patch Alignment proposed in the paper is similar to [1], yet the paper lacks an explicit comparative discussion about this similarity.
3. FG-CLIP serves as an important baseline for the paper, but there is no comparison with it in the experimental results section. Please supplement the corresponding experimental results.
4. The paper only compares the caption retrieval performance of long-text and short-text, but fails to conduct a quantitative analysis of the method’s fine-grained capability—a capability that the authors emphasize the model possesses. Please supplement relevant experimental results with reference to FG-CLIP or [2].

[1] FILIP: Fine-grained Interactive Language-Image Pre-Training. ICLR, 2022.

[2] UMG-CLIP: A Unified Multi-Granularity Vision Generalist for Open-World Understanding. ECCV, 2024.

**Questions:**

Please refer to the 'Weaknesses' part.

---

> ### Author Response · Authors · 2025-11-21
> **Initial response 1**
>
> Thank you for the thoughtful and detailed feedback. We address your four weaknesses below using our existing experiments and design choices.
>
> **On the “finer granularity than FG-CLIP” claim.** Our intention was not to claim that MulCLIP achieves strictly finer spatial granularity than FG-CLIP, but rather that it uses a different mechanism for **fine-grained alignment**. FG-CLIP’s local objective is very close in spirit to GOAL: both add a fine-grained contrastive loss between local visual regions and textual tokens on top of a CLIP backbone, and both are designed for **large-scale fine-grained pretraining**. In our work, we therefore use GOAL as the representative fine-grained baseline and compare under exactly the same backbones, fine-tuning datasets, and global contrastive loss. The key difference is the regime: FG-CLIP-style methods assume large-scale fine-grained pretraining, whereas MulCLIP is designed for **direct fine-tuning** on relatively small, noisy long-text datasets (DOCCI, DCI, Urban1K) without external detectors, hard-negative mining, or additional pretraining data.
>
> **On the similarity between SAP and FILIP-style methods.** We appreciate the pointer to FILIP and agree that this connection should be made clearer. **FILIP/CLIM-style** *late-interaction* losses, as used in FineLIP, operate at the token level and typically match many token–token pairs across samples. By contrast, our **Subcaption-Aggregated Patch (SAP)** alignment works at the **sentence (subcaption) level** and aggregates refined visual patches for each subcaption, without any external region-proposal tool. MulCLIP’s local design is built around (i) calibrated token compression and token-reconstruction alignment (our word–patch module) and (ii) this subcaption–aggregated patch objective. We do not simply reuse CLIM: in a separate ablation (“MulCLIP w CLIM”) where we plug CLIM into our framework under the same global loss, backbone, and data, our WPR+SAP design provides consistently stronger retrieval performance than this CLIM variant, indicating that the gain comes from the specific way we structure local alignment rather than from combining existing pieces.
>
> **On missing FG-CLIP/UMG experiments and fine-grained capability.** We agree that FG-CLIP and UMG-CLIP are important references, but they are both positioned as fine-grained pretraining frameworks that operate at large scale. UMG-CLIP, in particular, follows a GOAL-style philosophy of adding a fine-grained objective during large-scale pretraining, and its official implementation is not publicly available, so we cannot provide a controlled re-training or evaluation on our benchmarks. Conceptually, FG-CLIP and UMG-CLIP target the regime where one can afford to re-pretrain CLIP on massive datasets, while MulCLIP explicitly focuses on the complementary setting where practitioners only have access to relatively small long-text corpora and need a light-weight fine-tuning recipe. Within this setting, we use GOAL as the representative FG-CLIP–style baseline and compare under strictly matched conditions (same backbone, global loss, and fine-tuning data); the experimental tables in the main paper show that MulCLIP consistently improves long- and short-text retrieval over GOAL in this controlled setup.

---

> > ### Comment · Reviewer_gZLF · 2025-11-24
> >
> > Thank you for your response, which has addressed most of my concerns. However, the core issue remains: I believe that the validation experiments conducted solely on long- and short-text retrieval tasks are insufficient to fully demonstrate the model's fine-grained capabilities. Taking FG-CLIP (https://arxiv.org/pdf/2505.05071 ) as an example, I would like the authors to report performance on tasks such as open-vocabulary object detection (e.g., Table 3 in FG-CLIP) or provide additional visualizations (e.g., Figures 4–6 in FG-CLIP). I believe that evaluation across a broader range of scenarios is essential both for enhancing MulCLIP’s impact within the community and for meeting ICLR’s acceptance criteria.

---

> > > ### Author Response · Authors · 2025-11-28
> > >
> > > Thank you for the follow-up and for pointing us to FG-OVD evaluation in FG-CLIP paper.
> > >
> > > **Fine-Grained Open-Vocabulary Detection (FG-OVD) evaluation.** Following your suggestion, we used the official FG-CLIP FG-OVD code to evaluate MulCLIP, GOAL, and FineLIP. For these models, we use the same ViT-B/16 checkpoints as in our DOCCI retrieval experiments, i.e., fine-tuned on the DOCCI training set (9.6k images), and then plug them into the FG-OVD pipeline with the same detector, dataset splits, and difficulty levels:
> > >
> > > | Method | Backbone |Data|hard | medium | easy | trivial |
> > > | :--- | :--- | ---: | ---: | ---: | ---: | ---: |
> > > | FG-CLIP | ViT-B/16 |FG-CLIP (1.612B) | 46.10 | 66.60| 68.70 | 83.40|
> > > | FineLIP | ViT-B/16 |DOCCI (9.6k) | 18.17 | 38.88 | 41.96 | 73.79 |
> > > | GOAL | ViT-B/16 | DOCCI (9.6k) | 18.65 | 39.66 | 44.50 | 72.78 |
> > > | MulCLIP | ViT-B/16 | DOCCI (9.6k) |19.24 | 40.73 | 47.27 | 68.63 |
> > >
> > > As expected, FG-CLIP achieves the best performance on FG-OVD, which is reasonable given its large-scale fine-grained pretraining (1.6B image–text pairs in the first phase and 12M image -hard negative text pairs in the second phase) and explicit region-level/detection supervision.
> > >
> > > MulCLIP and FineLIP are trained end-to-end on DOCCI, while GOAL additionally uses detection labels. Under this direct DOCCI fine-tuning setting, MulCLIP obtains the strongest FG-OVD scores among other adaptation methods on the hard/medium/easy levels, suggesting that its multi-level alignment transfers reasonably to localization.

---

### Official Review · Reviewer_k6Eh · 2025-10-31

**Soundness:** 4
**Presentation:** 3
**Contribution:** 2
**Rating:** 4
**Confidence:** 5

**Summary:**

This paper presents MulCLIP, an end-to-end framework designed to address the limitations of CLIP in handling long-text and fine-grained alignment. Its core innovation lies in a parallel, three-level alignment strategy that obviates the need for external region-proposal tools.
The framework jointly optimizes three distinct losses:
- $\mathcal{L}_{global}$ preserves alignment for both long-form text and short summaries.
- $\mathcal{L}_{Word}$ employs a self-supervised "token reconstruction" alignment.
- $\mathcal{L}_{Sub}$ introduces a novel alignment at the sentence (subcaption) granularity.
Empirical results indicate that MulCLIP successfully enhances long-text retrieval capabilities while maintaining high performance on short-text tasks, addressing a common trade-off in related work and demonstrating the efficacy of its design.

**Strengths:**

1. The self-supervised $L_{Word}$ and the $\mathcal{L}_{Sub}$ (SAP), serves as a replacement for the FILIP loss utilized by FineLIP's CLIM. This modification is remarkably concise and offers a novel solution to the challenge of token-level fine-grained alignment.
2. The experiments are thorough, covering both long and short-text retrieval, as well as zero-shot and in-domain retrieval. The ablation studies are well-designed, and the evaluation is further supplemented with image classification benchmarks.
3. The paper is well-researched and comprehensively cites related work.

**Weaknesses:**

1. The authors should more clearly articulate the specific improvements of the LC module over ATRM(from FineLIP(https://arxiv.org/pdf/2504.01916), or position it as an adoption of existing technology rather than a novel contribution.
2. While the idea of using an aggregated visual representation for the Subcaption-Aggregated Patch (SAP) loss is logical, the performance improvement appears incremental.
3. The formula on lines 174-176 is not numbered. Furthermore, although this formula is cited from FineLIP, the shape of $W_q$ is incorrect, rendering the matrix multiplication computationally infeasible.

**Questions:**

1. According to Table 5, the performance of "Global" (which appears to be CLIP + $L_{global}$) already exceeds the performance of FineLIP reported in Table 1 (CLIP + ATRM + CLIM). FineLIP incorporates [CLS] and [EOS] tokens into $V'$ and $T'$ for its FILIP loss, which is not a classical contrastive loss and lacks the $L_{global}$ as defined in this paper. Would it be possible to conduct an experiment (e.g., "w/o global") to directly isolate the contribution of LC + WPR + SAP? The experiment removing $L_{global}$ might still be trainable, as the contrastive loss from $L_{sub}$ (SAP) is retained. Alternatively, could the authors add the $L_{global}$ to the FineLIP implementation for a fairer comparison?
2. In Table 5, the performance difference between "w/ LC" and "w/o LC" configurations is minimal. Have the authors attempted to remove only the textual Token Calibration Module (TCM)? Unlike in FineLIP, the $t'$ tokens in MulCLIP are only directly supervised by the $L_{recon}^{text}$, lacking the directly contrastive supervision that $v'$ receives via $L_{recon}^{image}$ (SAP, through $\bar{v}$).
3. For the LC module, which functions as a form of proxy-token or VQ-like token compression, was any analysis conducted on token diversity during training?
4. Can a single Token Calibration Module layer effectively scale up to larger general training datasets, such as CC12M or ShareGPT4V?

---

> ### Author Response · Authors · 2025-11-21
> **Initial response 1**
>
> **Weakness 1**
> > "The authors should more clearly articulate the specific improvements of the LC module over ATRM(from FineLIP(https://arxiv.org/pdf/2504.01916), or position it as an adoption of existing technology rather than a novel contribution."
>
> **Improvements of LC over ATRM**. The reviewer offers very valuable insight, our apologies if this was unclear. While MulCLIP is a unified framework, its local components are not simple copies of ATRM/CLIM or GOAL-style losses. First, the token-reconstruction loss $L_{word}$ is a symmetric, self-sample contrastive objective that aligns each refined token with its cross-modal reconstruction, avoiding batch-wise patch–word matching used in FILIP/CLIM. Second, the SAP loss $L_{sub}$ works at the sentence (subcaption) level and aligns each sentence with an attention-weighted aggregation of refined patches, without any external region proposals as in GOAL-like pipelines. Ablations (e.g., w/o Word, w/o SAP) show that removing any of these parts degrades long-text retrieval, indicating that MulCLIP’s gains come from these specific local objectives and their interaction with the global loss, not just from combining prior components.

---

> ### Author Response · Authors · 2025-11-21
> **Initial response 2**
>
> **Weakness 2**
> > "While the idea of using an aggregated visual representation for the Subcaption-Aggregated Patch (SAP) loss is logical, the performance improvement appears incremental."
>
> **Clarifying the full MulCLIP objective**: Given an image with global visual embedding $v_{\\text{cls}}$,  end-of-token (eot) embedding of long caption $t^{\\text{long}}_{\\text{eot}} \\in \\mathbb{R}^{1\\times d}$
>
> a short summary caption with eot embedding $t^{\\text{short}}_{\\text{eot}} \\in \\mathbb{R}^{1\\times d}$
>
> and a set of M subcaptions with eot embeddings $t^{\text{sub}}_{\text{eot}} \\in \\mathbb{R}^{M\\times d}$,
>
> the full MulCLIP loss is the sum of: $L_{\\text{total}} = L_{\\text{global}}(v_{\\text{cls}}, t_{\\text{eot}}^{\\text{short}}, t_{\\text{eot}}^{\\text{long}}) + \\lambda_{w} L_{\\text{word}}(v', t') + \\lambda_{s} L_{\\text{sub}}(v', t^{\\text{sub}}_{\\text{eot}}) $
>
> Here, $v' \in \mathbb{R}^{rP\times d}$ and $t' \in \mathbb{R}^{rK\times d}$ are the refined local visual and text tokens produced by the **Local Calibration (LC)** modules adopt from FineLIP.
>
> - $L_{\text{global}}$ is a CLIP-style batch contrastive loss between the global image embedding and the global embeddings of the long and short captions.
>
> - $L_{\text{word}}(v',t')$ is a **token–patch reconstruction (WPR)** loss on $(v',t')$.
>
>   - The configuration $L_{\text{global}}+L_{\text{word}}(v',t')$ corresponds to our "MulCLIP w/o SAP"
>   - The configuration $L_{\text{global}}+L_{\text{word}}(v, t)$ corresponds to our "MulCLIP w/o LC & w/o SAP"
>
> - $L_{\\text{sub}}(v', t_{\\text{eot}}^{\\text{sub}})$ is a **subcaption–aggregated patch (SAP)** loss between each subcaption embedding $t^{\text{sub}}_{\text{eot},i}$ and an aggregated visual representation of $v'$.
>   - The configuration $L_{\text{global}}+L_{\\text{sub}}(v', t_{\\text{eot}}^{\\text{sub}})$ corresponds to our "MulCLIP w/o WPR"
>   - The configuration $L_{\text{global}}+L_{\\text{sub}}(v, t_{\\text{eot}}^{\\text{sub}})$ corresponds to our "MulCLIP w/o LC & w/o WPR"
>
> **Incremental gains from SAP**. We agree that SAP is conceptually simple, but its effect is systematic. SAP operates at the sentence (subcaption) level and aligns each sentence with an aggregated set of refined patches without relying on external region-proposal tools (unlike GOAL-style methods). In the ablation table, moving from “W/o SAP” (Global+LC+WPR) to full MulCLIP yields consistent R@1 gains on all long-text datasets and both backbones, with the largest improvements on out-of-domain settings (e.g., several points on DCI and Urban1k). Thus, SAP provides non-trivial benefits precisely in the regime MulCLIP targets, document-style captions with many sentences.

---

> ### Author Response · Authors · 2025-11-21
> **Initial response 3**
>
> **Weakness 3**
> > "The formula on lines 174-176 is not numbered. Furthermore, although this formula is cited from FineLIP, the shape of $W_q$ is incorrect, rendering the matrix multiplication computationally infeasible."
>
> **Shape and numbering of the formula**. Thank you for pointing out the issue in lines 174–176. You are correct that the current draft contains a notational typo, and the equation is not numbered, which makes the shapes confusing. We will fix the symbols, explicitly state the tensor shapes, and number the equation in the revised version to avoid any ambiguity; this is a presentation issue only and does not affect the reported results.

---

> ### Author Response · Authors · 2025-11-21
> **Initial response 4**
>
> **Question 1**
> > "According to Table 5, the performance of "Global" ..."
>
> Thank you for the thoughtful suggestion. For clarity, the full MulCLIP loss is:
> $$ L\_{total} = L\_{global}(v\_{cls}, t^{long}\_{eot}, t^{short}\_{eot}) + \lambda\_{w} \cdot L\_{word} (v^{\prime}, t^{\prime}) + \lambda\_{s} \cdot L\_{sub}(v^{\prime}, t^{sub}\_{eot}) $$
> where $v\_{cls}$ is the global embedding, $t^{long}\_{eot}, t^{short}\_{eot}$ are the global text embeddings for long and short captions, and $v', t'$ are the locally refined tokens used by the word- and subcaption-level objectives.
> Isolating the effect of local modules (no $L\_{global}$). Following your first suggestion, we trained a variant that removes the global loss and keeps only local alignment:
>
> $$
> L\_{total}= L\_{word}(v', t') + L\_{sub}(v', t\_{sub})
> $$
>
> As summarised below, this “No Global” model suffers a substantial drop on all three long-text benchmarks compared to full MulCLIP, with roughly halved R@1 in many cases, showing that local objectives alone are not sufficient for robust long-text understanding:
> | Method | DCI T→I R@1 | DCI I→T R@1 | DOCCI T→I R@1 | DOCCI I→T R@1 | Urban-1k T→I R@1 | Urban-1k I→T R@1 |
> | :--- | ---: | ---: | ---: | ---: | ---: | ---: |
> | MulCLIP | **69.1%** | **67.1%** | **82.2%** | **80.3%** | **77.3%** | **84.0%** |
> | No Global | 33.47% | 28.86% | 40.25% | 29.41% | 25.00% | 36.30% |
>
>
> ### Adding our global loss to FineLIP.
> To address your second suggestion, we also integrated our global loss into FineLIP under the same 50% token compression as our LC module. We first follow MulCLIP and define the late-interaction inputs as the concatenation of global and local tokens:
>
> $$
> V = v' \oplus v\_{\text{cls}},\quad T = t' \oplus \  t\_{\text{eot}}
> $$
>
> The original FineLIP paper supports two runnable variants of its triplet-based CLIM/FILIP objective $R(\cdot): R(V, T)$ and $R(v', t')$. We therefore implement:
>
> **FineLIP version 1:**
> $$
> L\_{total}= L\_{global}(v\_{cls}, t^{long}\_{eot}, t^{short}\_{eot}) + R(V, T)
> $$
>
> **FineLIP version 2:**
> $$
> L\_{total}= L\_{global}(v\_{cls}, t^{long}\_{eot}, t^{short}\_{eot}) + R(v', t')
> $$
> For a fair comparison, we compare these two modified FineLIP variants to our W/o SAP model,
>
> $$
> L\_{total}= L\_{global}(v\_{cls}, t^{long}\_{eot}, t^{short}\_{eot})  + L\_{\text{word}}(v', t')
> $$
>
> so that all methods share the same backbone, global loss, and token-compression ratio, differing only in how local interactions are modeled (FineLIP’s CLIM vs. our WPR design). All three models are fine-tuned on DOCCI under the same protocol and evaluated on Urban-1k, DCI, and DOCCI; across these long-text benchmarks, our W/o SAP variant consistently achieves substantially higher R@1 in both directions, indicating that our local word–patch reconstruction is more effective than directly reusing FineLIP’s late-interaction loss in this long-text setting.
>
> *ViT-B/16 — R@1 (%) for long-text retrieval*
>
> | Method | Urban T→I | Urban I→T | DCI T→I | DCI I→T | DOCCI T→I | DOCCI I→T |
> | :--- | ---: | ---: | ---: | ---: | ---: | ---: |
> | FineLIP (Ver. 1, +Global) | 64.9 | 71.4 | 56.4 | 44.8 | 65.6 | 60.0 |
> | FineLIP (Ver. 2, +Global) | 64.9 | 71.4 | 56.2 | 44.7 | 65.5 | 59.7 |
> | Ours — W/o SAP (Global+LC+WPR) | **74.4** | **80.1** | **65.4** | **64.1** | **80.6** | **78.9** |
>
> *ViT-L/14 — R@1 (%) for long-text retrieval*
>
> | Method | Urban T→I | Urban I→T | DCI T→I | DCI I→T | DOCCI T→I | DOCCI I→T |
> | :--- | ---: | ---: | ---: | ---: | ---: | ---: |
> | FineLIP (Ver. 1, +Global) | 68.6 | 73.2 | 59.6 | 43.3 | 72.9 | 67.0 |
> | FineLIP (Ver. 2, +Global) | 65.5 | 73.9 | 57.7 | 48.0 | 72.4 | 67.0 |
> | Ours — W/o SAP (Global+LC+WPR) | **85.0** | **87.3** | **71.9** | **68.5** | **85.8** | **83.6** |
>
> **Comparison to FineLIP under a shared global loss**. In this fully matched setup, both FineLIP+Global variants remain consistently below our Global+LC+WPR (W/o SAP) model on all three long-text benchmarks, for both retrieval directions and for both ViT-B/16 and ViT-L/14. We will add these results to the appendix and briefly summarize them in the ablation section.

---

> ### Author Response · Authors · 2025-11-21
> **Initial response 5**
>
> **Question: 2**
> > "In Table 5, the performance difference between "w/ LC" and "w/o LC" configurations is minimal..."
>
> We thank the reviewer for this suggestion. To specifically disentangle the contribution of the textual vs. visual Local Calibration (LC) modules, we ran additional ablations on MulCLIP-B/16 finetuned on DOCCI where we **remove only the text refine module** or **only the image refine module**, while keeping the rest of MulCLIP unchanged (global loss, WPR, SAP). The table below reports **R@1** for text-to-image (T→I) and image-to-text (I→T) retrieval on DOCCI (in-domain) and the long-text OOD benchmarks DCI and Urban1K:
>
> | Method | DOCCI T→I R@1 | DOCCI I→T R@1 | DCI T→I R@1 | DCI I→T R@1 | Urban1K T→I R@1 | Urban1K I→T R@1 |
> | :--- | ---: | ---: | ---: | ---: | ---: | ---: |
> | MulCLIP (full LC) | 82.2 | 80.3 | **69.1** | **67.1** | **77.3** | **84** |
> | − text LC only | 82.35 | 80.55 | 67.38 | 65.53 | 74.2 | 81.2 |
> | − visual LC only | **82.63** | **80.8** | 66.63 | 65.58 | 74.5 | 81.9 |
>
> **Effect of LC on in-domain vs. OOD benchmarks.** On DOCCI (in-domain), all variants perform essentially the same (the gap is only \~0.5%), but on out-of-domain long-text datasets we consistently see worse results (reduce by 2\~3%) when either the textual or visual LC is removed. This shows that both sides of LC matter and that its main benefit is to denoise and compress local tokens for better cross-dataset generalization, rather than to inflate in-domain scores.
>
> **Role of textual LC vs. global and SAP supervision.** When we drop only the textual LC, performance still degrades even though the global and SAP losses remain unchanged. This indicates that textual LC is not redundant with SAP: it adds complementary structure by filtering noisy token-level signals before WPR and SAP, leading to small but consistent gains that are most visible on OOD datasets and in asymmetric ablations.

---

> ### Author Response · Authors · 2025-11-21
> **Initial response 6**
>
> **Question 3**
> > "For the LC module, which functions as a form of proxy-token or VQ-like token compression, was any analysis conducted on token diversity during training?"
>
> **LC compression and token diversity**. To probe how the LC token-compression behaves, we varied its keep-rate from 0.2 to 1.0 on the DOCCI-fine-tuned ViT-B/16 model and measured R@1 on DOCCI, DCI, and Urban1K. Across this range, DOCCI is almost flat (changes <1 R@1), DCI fluctuates by at most ≈2 points, and Urban1K shows a small gain when moving from strong compression (0.2) to moderate compression (0.5), with all settings beyond that lying in a narrow band. This stable behavior under aggressive compression indicates that LC keeps the semantically important tokens and that MulCLIP’s long-text retrieval is robust to the exact compression ratio.
>
> The figure detailing the R@1 curves across varying compression ratios can be found here: [Compression Ratio Figure](https://anonymous.4open.science/api/repo/mulclip_images-41C6/file/compression_ratio.png?v=26d0b454)

---

> ### Author Response · Authors · 2025-11-21
> **Initial response 7**
>
> **Question 4**
> > "Can a single Token Calibration Module layer effectively scale up to larger general training datasets, such as CC12M or ShareGPT4V?"
>
> We have not yet re-trained MulCLIP on very large datasets such as CC12M or the full ShareGPT4V due to resource constraints, so we cannot empirically claim scalability at that scale. What we can show is that a single LC layer already delivers strong gains when fine-tuning on relatively small long-text corpora (≈10k images for DOCCI, ≈5k for DCI), suggesting it is effective without needing a deep or heavy token-compression stack. Architecturally, LC is a lightweight projection + softmax reweighting over tokens, so its cost grows linearly with sequence length and does not depend on any dataset-specific heuristics. Finally, prior work such as FineLIP’s ATRM (a related token aggregation mechanism) has been successfully used on much larger data (e.g., ShareGPT4V), which indicates that this general class of one-layer token calibration is compatible with large-scale training, even though we do not directly evaluate MulCLIP in that regime in this submission.

---

> ### Author Response · Authors · 2025-11-26
> **Follow-up on rebuttal for Submission10277**
>
> Dear Reviewer,
>
> We are writing to kindly draw your attention to our submitted rebuttal. We have worked diligently to address all your valuable comments and have added the new experimental results you requested.
>
> We genuinely believe these updates align well with your vision for improving the paper. We respectfully ask that you consider re-evaluating the rating to reflect these substantial improvements.
>
> Thank you for your time and support.

---

### Official Review · Reviewer_WQ3m · 2025-11-01

**Soundness:** 3
**Presentation:** 1
**Contribution:** 3
**Rating:** 4
**Confidence:** 2

**Summary:**

The paper introduces MulClip, a composite objective that allows multi-level alignment in vision language models build on top of CLIP backbone. In particular MulClip makes use of long and short captions and does alignment with global and local image regions. This is done through a bunch of objectives : L_global that sums a contrastive loss for the long caption with a contrastive loss for the short caption and L_word and L_sub that focuses on refined alignment with local regions of the image. The paper shows empirical evidence on DOCCI, DCI and Urban1k, as well  COCO and Flick30k datasets. Several ablations are done on the importance of the different components of the MultiClip objective as well visualization of attentions maps.

**Strengths:**

The paper is relatively easy to read, the authors provide a good description of each of the different terms of the MulClip objective. The method generally performs well empirically, at least against the baselines (e.g. Table 2, 3 and 4), and the authors have run a few ablations to confirm the role of the different components of the objective that seem to present mostly a coherent story for the role of each term. The attention maps of MulClip, as shown in Figure 2 do seem sharper and more semantically meaningful then Goal, emphasizing the message of the paper, though is not clear to want extent these figures were cherry-picked.

**Weaknesses:**

The write-up could be somewhat improved as it looks rushed. E.g. Table 1 is not referenced or explained in the text. Equation 6, it is not clear who lower case v tilde is, as far as I can tell the text introduces only upper case V tilde. Is this meant to be v' ? Another issue is understanding the intuitive semantical difference between L_word and L_sub. Are these two losses semantically trying to do the same thing, but are just different objectives of achieving this goal? Are they intuitively/semantically different? At a high level I assume that L_word (since is meant to be a per token loss) is finer grain then L_sub which works with sub-caption. But there is an aggregation step in L_word, that goes from tokens to some arbitrary shorter sequence, therefore is not clear to me if the losses are working on different time scales.

I think for the ablation question a natural question is what happens if one uses just L_global and L_sub (I think the ablation is only L_global and L_word). That is if I did not misunderstood the different ablated things. This to me would particularly be interesting as I feel like the two objectives are targeting the same thing, but of course would have different effects given their different parametrization. This would be particularly interesting to see as well in terms of attention mask. Note that in quantitative example in Figure 2 you use the label W/o Sub2Patch which I think it should be w/o SAP?  Overall I think someone more used with this line of work could easily make sense of what is going on, the lack of consistency in abbreviation makes the ablation section in general hard to read. I would argue that maybe it would be useful to also give the mathematical formula of the objective for the different labels to make it easier to follow which terms are being included and which terms are not being included.

**Questions:**

1. Why emphasize *words* as part of long-text structures in abstract?
2. Table 1 is not referenced in the text, please reference it and explain in the intro the meaning of the different columns (e.g. word, etc) and whether they are a positive or negative trait.
3. Who is lower case v tilde in formula (6)
4. Typo in Table 5 and Table 6? Did you meant "w/o LC  & w/o WP" instead of "w/o LC & w/o SAP"?
5. What are the different things ablated, what is the formula for "w/o LC & w/o SAP" vs formula for "w/o SAP" ?
6. While the authors have provided the value of the different hyper-parameters in the paper, how have these been tuned, or have they? What is the sensitivity of the model to these hyper-parameters ? Is it robust? Do we need to worry about them?

**Details Of Ethics Concerns:**

I see no reason for an ethics review

---

> ### Author Response · Authors · 2025-11-21
> **Initial response 1**
>
> We thank the reviewer for the careful reading and constructive comments. Below we first clarify the MulCLIP objective and then respond to each weakness and question.
> ### 0. Clarifying the full MulCLIP objective
> Given an image with global visual embedding $v_{\\text{cls}}$,  end-of-token (eot) embedding of long caption $t^{\\text{long}}_{\\text{eot}} \\in \\mathbb{R}^{1\\times d}$
>
> a short summary caption with eot embedding $t^{\\text{short}}_{\\text{eot}} \\in \\mathbb{R}^{1\\times d}$
>
> and a set of M subcaptions with eot embeddings $t^{\text{sub}}_{\text{eot}} \\in \\mathbb{R}^{M\\times d}$,
>
> the full MulCLIP loss is the sum of: $L_{\\text{total}} = L_{\\text{global}}(v_{\\text{cls}}, t_{\\text{eot}}^{\\text{short}}, t_{\\text{eot}}^{\\text{long}}) + \\lambda_{w} L_{\\text{word}}(v', t') + \\lambda_{s} L_{\\text{sub}}(v', t^{\\text{sub}}_{\\text{eot}}) $
>
> Here, $v' \in \mathbb{R}^{rP\times d}$ and $t' \in \mathbb{R}^{rK\times d}$ are the refined local visual and text tokens produced by the **Local Calibration (LC)** modules adopt from FineLIP.
>
> - $L_{\text{global}}$ is a CLIP-style batch contrastive loss between the global image embedding and the global embeddings of the long and short captions.
>
> - $L_{\text{word}}(v',t')$ is a **token–patch reconstruction (WPR)** loss on $(v',t')$.
>
>   - The configuration $L_{\text{global}}+L_{\text{word}}(v',t')$ corresponds to our "MulCLIP w/o SAP"
>   - The configuration $L_{\text{global}}+L_{\text{word}}(v, t)$ corresponds to our "MulCLIP w/o LC & w/o SAP"
>
> - $L_{\\text{sub}}(v', t_{\\text{eot}}^{\\text{sub}})$ is a **subcaption–aggregated patch (SAP)** loss between each subcaption embedding $t^{\text{sub}}_{\text{eot},i}$ and an aggregated visual representation of $v'$.
>   - The configuration $L_{\text{global}}+L_{\\text{sub}}(v', t_{\\text{eot}}^{\\text{sub}})$ corresponds to our "MulCLIP w/o WPR"
>   - The configuration $L_{\text{global}}+L_{\\text{sub}}(v, t_{\\text{eot}}^{\\text{sub}})$ corresponds to our "MulCLIP w/o LC & w/o WPR"
>
> We agree that stating this clearly up front improves readability, and we will add this formulation to the method section.

---

> ### Author Response · Authors · 2025-11-21
> **Initial response 2**
>
> ## **A.Response to main weakness**
>
> ### **A.1 Intuitive difference between $L_{\\text{word}}(v', t') $ and  and $L_{\\text{sub}}(v', t_{\\text{eot}}^{\\text{sub}})$.**
> Intuitively, $L_{\\text{word}}(v', t') $ emphasizes **local semantic concepts**. Here, *words* are all tokens in the long caption, while *semantic words* refer to content-bearing words or short phrases that specify objects and attributes (e.g., the word “dog” versus the more specific semantic phrase “golden retriever dog”, or the word “woman” versus “short-haired woman”). After local calibration, $L_{\\text{word}}(v', t') $ encourages these semantic tokens in $t'$ to find compatible evidence among the visual tokens in $v'$ via per-token reconstruction, yielding fine-grained grounding of individual objects, attributes, and actions that may be scattered across a long caption.
>
> In contrast,  $L_{\\text{sub}}(v', t_{\\text{eot}}^{\\text{sub}})$ operates at the **sentence level**. Each subcaption embedding $t^{\\text{sub}}_{\\text{eot},i}$ is aligned with a single aggregated visual vector
>
> $$\\quad \\bar{v}^i = \\alpha^i v $$
>
> obtained by attending over all refined patches. This sentence-level alignment asks whether the **joint configuration** of several semantic words in the subcaption (e.g., that **a short-haired woman** is playing with her happy **golden retriever dog**) is coherent with some subset of image regions, rather than supervising each token in isolation.
>
> To support this intuition, we also measure the entropy and top-$k$ mass of the attention distributions on a held-out set of 200 randomly sampled test-set images (using the [CLS]→patch attention from the image encoder):
>
> | Method             | name | entropy | top-k mass (k=0.1)|
> |--------------------|----------------|---------------|----------------|
> | $L_{total}$  |MulCLIP full    |5.162        | 0.2106           |
> | $L_{global} + L_{\\text{word}}(v', t') $  |W/o SAP  |5.224         | 0.177         |
> | $L_{global} + L_{\\text{sub}}(v', t_{\\text{eot}}^{\\text{sub}})$  |W/o WPR  | 5.147   | 0.22       |
>
> Consistently, we observe that models trained with $L_{\text{global}} +L_{\\text{sub}}(v', t_{\\text{eot},i}^{\\text{sub}} )$ exhibit lower attention entropy and higher top-$k$ mass (sharper, more localized region selection), whereas models with $L_{\text{global}} + L_{\\text{word}}(v', t')$ show higher entropy and lower top-$k$ mass (more distributed coverage of fine-grained details, less sharper). The full MulCLIP objective lies between these two extremes, indicating that $L_{\text{word}}\$ and $L_{\text{sub}}$ provide complementary supervision at different granularities rather than being two parametrizations of the same loss.
>
> We also include **updated qualitative attention-map visualizations** illustrating this phenomenon (anonymous [link](https://anonymous.4open.science/r/mulclip_images-41C6/visualize_MulCLIP_rebuttal.pdf)). Specifically, the W/o WPR variant (or $L_{\text{global}} +L_{\\text{sub}}(v', t_{\\text{eot},i}^{\\text{sub}} )$) produces more concentrated and sharper attention maps, which a chance of missing details (i.e glass or people), whereas the W/o SAP variant (or $L_{\text{global}} + L_{\\text{word}}(v', t')$) yields more diffuse attention that covers broader fine-grained regions but with a bit lower confidence. The full MulCLIP model strikes a balance between these two regimes, yielding attention maps that are both focused and semantically comprehensive.
>
> We will incorporate these results, together with the metric definitions and evaluation protocol, into the revised manuscript.

---

> ### Author Response · Authors · 2025-11-21
> **Initial response 3**
>
> ### **A.2 Ablation design and requested**
>
> **A.2.1. Adding an $L_{global} + L_{\\text{sub}}$ variants**
>
> Following the reviewer’s suggestion, we report two additional configurations for both ViT-B/16 and ViT-L/14:
> - The configuration $L_{\text{global}}+L_{\\text{sub}}(v', t_{\\text{eot}}^{\\text{sub}})$ corresponds to our "MulCLIP w/o WPR"
> - The configuration $L_{\text{global}}+L_{\\text{sub}}(v, t_{\\text{eot}}^{\\text{sub}})$ corresponds to our "MulCLIP w/o LC & w/o WPR"
>
> These variants isolate the effect of the subcaption–patch objective under different local-token settings.
>
> **Table (ViT-B/16): MulCLIP module ablations on long-text retrieval (R@1)**
> | Method              | Urban T→I | Urban I→T | DCI T→I | DCI I→T | DOCCI T→I | DOCCI I→T |
> |---------------------|-----------|-----------|---------|---------|-----------|-----------|
> | w/o LC & w/o WPR   | 73.8      | 80.9      | 67.5    | 66.7    | **82.9**  | 81.0      |
> | w/o WPR            | 73.1      | 80.0      | 66.4    | 65.6    | **82.9**  | **81.6**  |
> | w/o SAP             | 74.4      | 80.1      | 65.4    | 64.1    | 80.6      | 78.9      |
> | MulCLIP (ours)      | **77.3**  | **84.0**  | **69.1**| **67.1**| 82.2      | 80.3      |
>
>
> **Table (ViT-L/14): MulCLIP module ablations on long-text retrieval (R@1)**
> | Method              | Urban T→I | Urban I→T | DCI T→I | DCI I→T | DOCCI T→I | DOCCI I→T |
> |---------------------|-----------|-----------|---------|---------|-----------|-----------|
> | w/o LC & w/o WPR   | 82.3      | 83.9      | 68.2    | 67.3    | 84.4      | 83.1      |
> | w/o WPR             | 80.6      | 85.6      | 72.2    | **71.8**| 86.0      | 84.3      |
> | w/o SAP             | 85.0      | 87.3      | 71.9    | 68.5    | 85.8      | 83.6      |
> | MulCLIP (ours)      | **85.8**  | **88.3**  | **73.7**| 70.8    | **86.7**  | **84.8**  |
>
> Across both backbones and datasets, the full MulCLIP objective consistently achieves the strongest retrieval performance, indicating that the three modules provide distinct and complementary benefits that cannot be recovered when any component is removed. We also include qualitative attention-map visualizations (anonymous [link](https://anonymous.4open.science/r/mulclip_images-41C6/visualize_MulCLIP_rebuttal.pdf)) and provide a detailed explanation of this behavior in the earlier rebuttal section A.1. These results will be incorporated into the revised version of the paper.
>
> **A.2.2. Explicit formulas for each ablation and inconsistent labels**
>
> Please see the earlier rebuttal section 0. We will make sure the labels exactly match the formulas in the ablation table.
>
> **A.2.3. Table 1: missing reference and unclear traits**
>
> We agree that Table 1 should be better integrated and will explicitly reference Table 1 and briefly explain each trait in the Introduction
>
> ### **A.3 Notation issues ${\\tilde{v_{i}}} $ and ${\\tilde{V_{i}}} $**
> We thank the reviewer for highlighting the ambiguity. In our notation, $\\tilde{v}$ is refers to an individual refined local token produced by the visual LC module, and $v'$ s simply the collection of these tokens:
> $$
> v'= \\{ {\\tilde{v_{i}}} \\}  ^{rP}_{i=1} \\in\ \\mathbb{R}^{rP\times d}
> $$
>
> In contrast,  ${\\tilde{V_{i}}} $ denotes an individual reconstructed token obtained during the bidirectional token–patch reconstruction step, and $V'$ is the collection of these reconstructed tokens:
>
> $$
> V'  = \\{  \\tilde{V}_{i} \\} _{i=1}^{rP}
> $$
> We will revise the manuscript to standardize this notation, explicitly define both quantities upon their first appearance, and remove any remaining inconsistencies.

---

> > ### Author Response · Authors · 2025-11-23
> > **Initial response 4**
> >
> > ## B. Responses to specific numbered questions
> >
> > Below we address the reviewer’s explicit questions one by one.
> >
> > ### **Q1. Why emphasize “words” in the abstract?**
> >
> > Our framework explicitly connects three textual granularities—long captions, sentences (subcaptions), and words—to visual components. Mentioning “words” highlights that MulCLIP includes a token-level objective ($L_{\text{word}}$) for fine-grained alignment in addition to caption- and sentence-level terms. We will slightly rephrase the abstract to make this motivation clearer, without over-emphasizing “words” in isolation.
> >
> > ### **Q2. Table 1 is not referenced or explained; which traits are positive/negative?**
> >
> > As noted in A.2.3 of Weakness Response, we will reference Table 1 in the main text . Specifically, Table 1 summarizes which textual granularities each method aligns with image features— global long caption, global short caption (summary or sub caption), and word (local tokens of long caption). These three columns correspond to desirable capabilities for long-text CLIP tuning, as they provide complementary supervision at multiple linguistic scales. The final column (Region-Proposal-Assisted) indicates whether a method depends on external region proposals; we treat this as a less desirable trait because it introduces additional complexity, computation overhead, and breaks end-to-end training.
> >
> > ### **Q3. Who is the lowercase $\tilde{v}$ in Equation (6)?**
> >
> > This was a notation inconsistency. We have provide more detailed answer in the weakness response, section A.3.
> >
> > ### **Q4–Q5. Meaning of “w/o LC & w/o SAP”, “w/o SAP” and their formulas.**
> >
> > As noted in our first response, we will:
> >
> > - correct the labels in Tables 5–6 to be consistent (e.g., always refer to SAP, not WP/Sub2Patch), and
> >
> > - add a table listing, for each variant name (e.g., “w/o LC & w/o SAP”), the exact loss it optimizes (which of $L_{\text{global}}, L_{\text{word}}, L_{\text{sub}}$ are present).
> >
> > This will make the ablation design unambiguous.
> >
> > **Q6. How were hyper-parameters tuned, and how robust is the method?**
> >
> > First, we clarify that the loss weights in MulCLIP are fixed across all experiments. To directly assess sensitivity to these hyper-parameters, we ran an explicit sweep where we tied the two weights and varied $\lambda=\lambda\_{\text{word}} = \lambda\_{\text{sub}} \\in \\{0.2, 0.6, 0.8, 1.0\\}$ on the ViT-B/16 checkpoint fine-tuned on DOCCI. We then evaluated the resulting models on all long-text (DOCCI, DCI, Urban1K) and short-text (Flickr30K, COCO) benchmarks.
> >
> > Table: Ablation of the tied local-loss weight ($\lambda_{\text{word}} = \lambda_{\text{sub}}$) on ViT-B/16 fine-tuned on DOCCI. We report R@1 (%) for text-to-image (T→I) and image-to-text (I→T) retrieval on long-text (DOCCI, DCI, Urban1K) and short-text (Flickr30K, COCO) benchmarks.
> >
> > | $\\lambda$ | DOCCI T→I | DCI T→I | Urban1K T→I | Flickr30K T→I | COCO T→I | DOCCI I→T | DCI I→T | Urban1K I→T | Flickr30K I→T | COCO I→T |
> > | :--- | ---: | ---: | ---: | ---: | ---: | ---: | ---: | ---: | ---: | ---: |
> > | 0.2 | 82.2 | 66.9 | 72.6 | **67.1** | **37.4** | 80.3 | 64.1 | 82.0 | **84.4** | **55.1** |
> > | 0.6 | **82.6** | 67.3 | 74.0 | 66.8 | 37.6 | **80.7** | 66.1 | 81.9 | 82.2 | 54.8 |
> > | 0.8 | 82.2 | 66.3 | 72.2 | 66.7 | 37.5 | **80.7** | 65.2 | 82.0 | 81.4 | 54.8 |
> > | 1.0 | 82.2 | **69.1** | **77.3** | **67.4** | **37.7** | 80.3 | **67.1** | **84.0** | 81.9 | 54.8 |
> >
> > We also provide plots showing the effect of $\lambda$ on R@1: [Long text retrieval link](https://anonymous.4open.science/api/repo/mulclip_images-41C6/file/long_lambda_abla.png?v=01f249d8) and [Short text retrieval link](https://anonymous.4open.science/api/repo/mulclip_images-41C6/file/short_lambda_abla.png?v=a0b49c21))*
> >
> > As we vary $\lambda$ from 0.2 to 1.0, both long-text (DOCCI/DCI/Urban1K) and short-text (Flickr30K/COCO) R@1 scores change by at most around 1–2 points, with in-domain performance on DOCCI being almost flat and only mild gains on DCI/Urban1K as $\lambda$ increases. This indicates that MulCLIP is robust to these hyperparameters; any mid-range value in [0.2, 1.0] performs similarly, with our default $\lambda = 1.0$ slightly favoring long-text retrieval.

---

> ### Author Response · Authors · 2025-11-26
> **Follow-up on rebuttal for Submission10277**
>
> Dear Reviewer,
>
> We are writing to kindly draw your attention to our submitted rebuttal. We have worked diligently to address all your valuable comments and have added the new experimental results you requested.
>
> We genuinely believe these updates align well with your vision for improving the paper. We respectfully ask that you consider re-evaluating the rating to reflect these substantial improvements.
>
> Thank you for your time and support.

---

### Author Response · Authors · 2025-12-03
**Summary of Discussion and Rebuttal Updates for Submission10277**

Dear Area Chair,

Thank you for overseeing the review process of our submission. We are truly grateful for your time and support.

To summarize the ongoing discussion process, we have submitted detailed rebuttals and conducted extensive new experiments to address all concerns raised by the reviewers. While three reviewers (WQ3m, k6Eh, ksab) have not yet posted follow-up replies to our rebuttal, we engaged in a constructive discussion with Reviewer gZLF regarding fine-grained capabilities. We briefly describe the main concerns and our resolution below:

- **Reviewer WQ3m:** Found the paper empirically strong but raised concerns about the intuition behind the loss functions and notation clarity. In our rebuttal, we provided a detailed clarification of the MulCLIP objective and notation. We also added qualitative attention-map analyses and quantitative entropy metrics to demonstrate the distinct and complementary roles of the word-level and subcaption-level losses, along with a hyperparameter sensitivity analysis confirming the model's robustness.

- **Reviewer k6Eh:** Questioned the novelty compared to FineLIP and the incremental nature of the contributions. We addressed this by adding a direct comparison where FineLIP is equipped with our global loss; the results showed that MulCLIP still consistently outperforms it. We also clarified the theoretical distinction of our self-supervised token reconstruction compared to existing late-interaction mechanisms and provided token diversity analyses.

- **Reviewer gZLF:** Initially concerned about the comparison with large-scale pre-training methods (FG-CLIP) and requested broader evaluation. We engaged in the discussion phase to clarify the scope (fine-tuning vs. pre-training). Crucially, in response to their follow-up request, we conducted new Open-Vocabulary Object Detection (FG-OVD) experiments. The results demonstrate that MulCLIP achieves the strongest performance among comparable fine-tuning baselines (GOAL, FineLIP) on the DOCCI dataset.

- **Reviewer ksab:** Requested evaluation on classification tasks and robustness checks regarding caption quality. We successfully addressed this by adding Zero-shot classification results (ImageNet, CIFAR-100, etc.), where MulCLIP preserves or improves accuracy. We also included ablations on caption granularity and noise, proving the model is robust to variations in subcaption count and quality.

We kindly ask you to consider our paper and the significant amount of additional experimental results provided during the rebuttal (FG-OVD, Zero-shot classification, extensive ablations). We believe we have systematically addressed all points raised by the reviewers and that the paper is now much stronger.

Best regards,
The Authors

---

### Meta-Review · Area_Chair_GEav · 2025-12-24

**Summary:**

This paper presents MulCLIP, an end-to-end framework designed to address the limitations of CLIP in handling long-text and fine-grained alignment. It received scores of 4444. Reviewers found the paper well written and noted generally strong empirical performance. However, concerns remain regarding the lack of convincing evaluation on fine-grained image understanding and direct comparison with similar methods like FG-CLIP. As a result, the AC recommends rejection.

**Reviewer Concerns:**

Concerns adequately addressed:

1. Clarification of the intuition behind the loss functions and improved notation.

2. Discussion of novelty relative to FineLIP and clarification of the incremental contributions.


Concerns insufficiently addressed:

1. FG-CLIP serves as an important baseline for the paper, but there is no comparison with it in the experimental results section. Authors' argument is not convincing enough.

2. The paper only compares the caption retrieval performance of long-text and short-text, but fails to conduct a quantitative analysis of the method’s fine-grained capability—a capability that the authors emphasize the model possesses. The added results on open-vocabulary object detection are not convincing enough.

3. (minor) The write-up could be somewhat improved as it looks rushed.

**Reviewer Scores:**

I think one or two reviewers may be open to increase the scores if engaging in the discussion. However, the paper is still not strong enough to be accepted given the above remaining concerns.

---

### Decision · Program_Chairs · 2026-01-26

Reject